

# A proxy for atmospheric daytime gaseous sulfuric acid
# concentration in urban Beijing
Yiqun Lu[1], Chao Yan[2], Yueyun Fu[3], Yan Chen[4], Yiliang Liu[1], Gan Yang[1], Yuwei Wang[1],
Federico Bianchi[2], Biwu Chu[2], Ying Zhou[5], Rujing Yin[3], Rima Baalbaki[2], Olga Garmash[2],
Chenjuan Deng[3], Weigang Wang[4], Yongchun Liu[5], Tuukka Petäjä[2,5,6], Veli-Matti Kerminen[2],
Jingkun Jiang[3], Markku Kulmala[2,5], Lin Wang[1,7,8*]
[1] Shanghai Key Laboratory of Atmospheric Particle Pollution and Prevention (LAP[3]),
Department of Environmental Science & Engineering, Jiangwan Campus, Fudan University,
Shanghai 200438, China
[2] Institute for Atmospheric and Earth System Research / Physics, Faculty of Science, University
of Helsinki, 00014 Helsinki, Finland
[3] State Key Joint Laboratory of Environment Simulation and Pollution Control, School of
Environment, Tsinghua University, Beijing 100084, China
[4] Institute of Chemistry, Chinese Academy of Sciences, Beijing 100190, China
[5] Aerosol and Haze Laboratory, Advanced Innovation Center for Soft Matter Science and
Engineering, Beijing University of Chemical Technology, Beijing 100029, China
[6] Joint International Research Laboratory of Atmospheric and Earth System Sciences
(JirLATEST), School of Atmospheric Sciences, Nanjing University, Nanjing 210023, China
[7] Institute of Atmospheric Sciences, Jiangwan Campus, Fudan University, Shanghai 200438,
China
[8] Shanghai Institute of Pollution Control and Ecological Security, Shanghai 200092, China
*Corresponding Author: L.W., email, lin_wang@fudan.edu.cn; phone, +86-21-31243568.
**Abstract**. Gaseous sulfuric acid is known as one of the key precursors for atmospheric
new particle formation processes, but its measurement remains a major challenge. A
proxy method that is able to derive gaseous sulfuric acid concentrations from
parameters that can be measured relatively easily and accurately is therefore highly
desirable among the atmospheric chemistry community. Although such methods are
available for clean atmospheric environments, a proxy that works well in a polluted
atmosphere, such as those in Chinese megacities, is yet to be developed. In this study,
the gaseous sulfuric acid concentration was measured in February-March, 2018, in
urban Beijing by a nitrate based - Long Time-of-Flight Chemical Ionization Mass
Spectrometer (LToF-CIMS). A number of atmospheric parameters were recorded
concurrently including the ultraviolet radiation B (UVB) intensity, concentrations of $O_3$,
$NO_x$, $SO_2$ and HONO, and aerosol particle number size distributions. A proxy for
atmospheric daytime gaseous sulfuric acid concentration was derived using a statistical



analysis method by using the UVB intensity, [$SO_2$], condensation sink (CS), [$O_3$], and
[HONO] (or [$NO_x$]) as the predictor variables. In this proxy method, we considered the
formation of gaseous sulfuric acid from reactions of $SO_2$ and OH radicals during the
daytime, and loss of gaseous sulfuric acid due to its condensation onto the pre-existing
particles. In addition, we explored formation of OH radicals from the conventional gas-
phase photochemistry using ozone as a proxy and from the photolysis of
heterogeneously-formed HONO using HONO (and subsequently $NO_x$) as a proxy. Our
results showed that the UVB intensity and [$SO_2$] are dominant factors for the production
of gaseous sulfuric acid, and that the simplest proxy could be constructed with the UVB
intensity and [$SO_2$] alone, resulting in up to 29% relative deviations when sulfuric acid
concentrations were larger than $2.0 \times 10^6$ molecules cm$^{-3}$. When the OH radical
production from both homogenously- and heterogeneously-formed precursors were
considered, the relative deviations were lower than 24%.



## 1 Introduction

Gaseous sulfuric acid ($H_2SO_4$) is a key precursor for atmospheric new particle formation (NPF) processes (Kerminen, 2018; Kirkby et al., 2011; Kuang et al., 2008; Kulmala and Kerminen, 2008; Sipilä et al., 2010). A number of atmospheric nucleation mechanisms including $H_2SO_4$-$H_2O$ binary nucleation (Benson et al., 2008; Duplissy et al., 2016; Kirkby et al., 2011), $H_2SO_4$-$NH_3$-$H_2O$ ternary nucleation (Kirkby et al., 2011; Korhonen et al., 1999; Kürten et al., 2015), and $H_2SO_4$-DMA-$H_2O$ ternary nucleation (Almeida et al., 2013; Petäjä et al., 2011; Yao et al., 2018) demand the participation of gaseous sulfuric acid molecules. In addition, the condensation of gaseous sulfuric acid onto newly-formed particles contributes to their initial growth (Kulmala et al., 2013). Quantitative assessments of the contribution of gaseous sulfuric acid to both the new particle formation rates and the particle growth rates require real-time measurements of gaseous sulfuric acid concentrations during the NPF events (Nieminen et al., 2010; Paasonen et al., 2010).

Measurements of gaseous sulfuric acid in the lower troposphere are challenging because its ambient concentration is typically quite low ($10^6$-$10^7$ molecule cm$^{-3}$) (Kerminen et al., 2010; Mikkonen et al., 2011). Reported real-time measurements of gaseous sulfuric acid are currently based on Chemical Ionization Mass Spectrometry with $NO_3^-$ as reagent ions (nitrate CIMS) because CIMS has a low detection limit for the atmospheric concentration range of gaseous sulfuric acid (Jokinen et al., 2012), and a constant fraction of sulfuric acid present in the air sample will be ionized by excessive nitrate ions in CIMS under constant instrumental conditions (Kürten et al., 2012; Zheng et al., 2010), which makes the quantification of gaseous sulfuric acid feasible.

Arnold and Fabian (1980) measured the negative ions in the stratosphere using a passive CIMS and derived the concentration of stratospheric gaseous sulfuric acid from the fractional abundances of a series of stratospheric negative ions as well as the associated equilibrium or rate constants. Later, real-time measurement of sulfuric acid in the lower troposphere was performed using nitrate CIMS (Eisele and Tanner, 1993), with laboratory calibrations by production of known concentrations of OH radicals that will be titrated into gaseous sulfuric acid. Thereafter, measurements of sulfuric acid using CIMS have been performed around the world (e.g., Berresheim et al., 2000; Bianchi et al., 2016; Jokinen et al., 2012; Kuang et al., 2008; Kürten et al., 2014; Kurtén et al., 2011; Petäjä et al., 2009; Weber et al., 1997; Zheng et al., 2011), and CIMS has been proved to be a robust tool for gaseous sulfuric acid detection. However, sulfuric





acid measurements are still rather sparse because of the high cost of the CIMS
instrument and the extensive demand of specialized expertise on the instrument
calibration, maintenance, and data processing, etc. Therefore, a proxy for gaseous
sulfuric acid concentration is highly desirable.
Proxies for the estimation of atmospheric gaseous sulfuric acid concentrations
were previously developed to approximate measurement results of sulfuric acid
in Hyytiälä, Southern Finland (Petäjä et al., 2009), supposing that gaseous sulfuric acid
is formed from reactions between $SO_2$ and OH radicals, and lost due to its condensation
onto pre-existing particles. The derived simplest proxy can be written as Eq. (1) below,
and the authors recognized that the proxies might be site-specific and should be verified
prior to utilization in other environments.

$$[\boldsymbol{H_2SO_4}] = \boldsymbol{k} \cdot \frac{[\boldsymbol{SO_2}] \cdot (\boldsymbol{UVB\ or\ Global\ radiation})}{\boldsymbol{CS}} \qquad (\boldsymbol{1})$$

Mikkonen et al. (2011) later developed a couple of statistical proxies based on
measurements of sulfuric acid in six European and North American sites, including
urban, rural and forest areas. Their results showed that the radiation intensity and $[SO_2]$
are the most important factors to determine the concentration of sulfuric acid, and that
the impact of condensation sink (CS), a proxy for condensational sink for gaseous
sulfuric acid, is generally negligible. In several proxies developed by Mikkonen et al.
(2011), the correlation between the gaseous sulfuric acid concentration and CS is
positive, which is against what one would expect. In addition, the performance of a
proxy equation is site-specific because of varying atmospheric conditions from one site
to another, which implies that the proxy suggested by Mikkonen et al. (2011) might not
work well in locations that characterized with an atmospheric environment different
from those in the six sites of that study.
Beijing is a location with typical values of CS being 10-100 times higher
(Herrmann et al., 2014; Wu et al., 2007; Xiao et al., 2015; Yue et al., 2009; Zhang et al.,
2011) and typical $SO_2$ concentrations being 1-10 times higher (Wang et al., 2011a; Wu
et al., 2017) than those in Europe and North America (Mikkonen et al., 2011), yet
measured gaseous sulfuric acid concentrations are relatively similar between these
environments (Wang et al., 2011b; Zheng et al., 2011). Whether previous proxies
developed for European and North American sites work in Beijing remains to be tested.





Furthermore, in addition to the gas phase reaction between $O(^1D)$ and water molecules
(Crutzen and Zimmermann, 1991; Logan et al., 1981), photolysis of HONO could be a
potentially important source of OH radical in the atmosphere not only in the early
morning (Alicke et al., 2002, 2003; Elshorbany et al., 2009; Li et al., 2012) but also
during the daytime (Acker et al., 2005; Aumont et al., 2003; Kleffmann, 2007). An
experimental study measuring HONO near the surface layer estimated that HONO was
a main contributor to OH production in Beijing, with HONO's contribution being larger
than 70% at around 12:00-13:00, except for summer when the contribution of $O_3$
dominated (Hendrick et al., 2014). Given the distinct characteristics of these two OH
radical formation pathways, they both should be included and evaluated separately
when a proxy for atmospheric gaseous sulfuric acid concentration is being built. The
reactions between $SO_2$ and criegee intermediates formed from the ozonolysis of
atmospheric alkenes could be a potential source of sulfuric acid only in the absence of
solar radiation (Boy et al., 2013; Mauldin et al., 2012), so these reactions are expected
to provide a minor contribution to the formation of gaseous sulfuric acid during the
daytime in urban Beijing.

In this study, gaseous sulfuric acid concentration was measured by a Long Time-

of-Flight Chemical Ionization Mass Spectrometer (LToF-CIMS) in February - March,
2018, in urban Beijing. A number of atmospheric parameters were recorded
concurrently, including the ultraviolet radiation B (UVB) intensity, concentrations of
$O_3$, $NO_x$, $SO_2$ and HONO, and particle number size distributions. The objective of this
study is to develop a robust daytime gaseous sulfuric acid concentration proxy for
Beijing, a representative Chinese megacity with urban atmospheric environments.

**2 Ambient measurements**

An intensive campaign was carried out from 9 February to 14 March, 2018 on the

fifth floor of a teaching building in the west campus of Beijing University of Chemical
Technology ($39°94´ N, 116°30´ E$). This monitoring site is 2 km to the west of the West
$3^{rd}$ Ring Road and surrounded by commercial properties and residential dwellings.
Hence, this station can be regarded as a representative urban site.

The sulfuric acid concentration was measured by a LToF-CIMS (Aerodyne

Research, Inc.) equipped with a nitrate chemical ionization source. Ambient air was
drawn into the ionization source through a stainless-steel tube with a length of 1.6 m
and a diameter of 3/4 inch. A mixture of a 3 standard cubic centimeter per minute (sccm)





ultrahigh purity nitrogen flow containing nitric acid and a 20 standard liter per minute
(slpm) pure air flow supplied by a zero-air generator (Aadco 737, USA), together as a
sheath flow, was guided through a PhotoIonizer (Model L9491, Hamamatsu, Japan) to
produce nitrate reagent ions. This sheath flow was then introduced into a co-axial
laminar flow reactor concentric to the sample flow. Nitrate ions were pushed to the
middle of sample flow under an electric field and subsequently charged sample
molecules. During the campaign, the sample flow rate was kept at 8.4 slpm, since mass
flow controllers fixed the sheath flow rate and the excess flow rate and the flow into
the mass spectrometer (around 0.4 slpm) was fixed by the size of a pinhole between the
ionization source and the mass spectrometer. The CIMS was calibrated twice during
the campaign following the protocols in previous literatures (Kürten et al., 2012; Zheng
et al., 2015). Here we use $1.1 \times 10^{10}$ molecule cm⁻³ as the calibration coefficient, after
taking into account diffusion losses in the stainless-steel tube and the nitrate chemical
ionization source. The obtained mass spectra were analyzed with a tofTools package
based on the MATLAB software (Junninen et al., 2010).

Ambient particle number size distributions down to about 1 nm were measured

using a combination of a scanning mobility particle sizer spectrometer (SMPS)
equipped with a diethylene glycol-based condensation particle counter (DEG-CPC, ~1-
10 nm) and a conventional particle size distribution system (PSD, ~3-700 nm)
consisting of a pair of aerosol mobility spectrometers developed by Tsinghua
University (Cai et al., 2017; Jiang et al., 2011; Liu et al., 2016). The values of CS were
calculated following Eq. (2) (Dal Maso et al., 2002):
$$CS = 2\pi D \int_0^\infty D_p \beta_m(D_p) n(D_p) dD_p = 2\pi D \sum_i \beta_i D_{pi} N_i \quad (2)$$
where $D_{pi}$ is the geometric mean diameter of particles in the size bin $i$ and $N_i$ is the
particle number concentration in the corresponding size bin. $D$ is the diffusion
coefficient of gaseous sulfuric acid, and $\beta_m$ represents a transition-regime correction
factor that could be defined as a function of the Knudsen number (Fuchs and Sutugin,

1971).

SO₂, O₃ and NOₓ concentrations were measured using a SO₂ analyzer (Model 43i,

Thermo, USA), a O₃ analyzer (Model 49i, Thermo, USA) and a NOₓ analyzer (Model
42i, Thermo, USA) with the detection limits of 0.1 ppbv, 0.5 ppbv and 0.4 ppbv,
respectively. The above instruments were pre-calibrated before the campaign. The UVB
(280 - 315 nm) intensity (UV-S-B-T, KIPP&ZONEN, The Netherlands) was measured



on the rooftop of the building. Atmospheric HONO concentrations were measured by
a home-made HONO analyzer with a detection limit of 0.01 ppbv (Tong et al., 2016).

Particle number size distributions and concentrations of gaseous sulfuric acid, $SO_2$,

$O_3$, $NO_2$ and HONO were recorded with a time resolution of 5 min, and the UVB
intensity with time resolution of 1 min. A linear interpolation method was used for
deriving the variables with the same time intervals, *i.e.*, 5 min. Only data between local
sunrise and sunset were used in the subsequent analysis.

**3 Development of a proxy for atmospheric gaseous sulfuric acid**

We derived the gaseous sulfuric acid concentration proxy on the basis of currently

accepted formation pathways of sulfuric acid in the atmosphere (R1-R3) (Finlayson-
Pitts and Pitts, 2000; Stockwell and Calvert, 1983):

$$OH + SO_2 \ \rightarrow \ HSO_3 \qquad\qquad (R1)$$
$$HSO_3 \ + \ O_2 \ \rightarrow \ SO_3 + HO_2 \qquad (R2)$$
$$SO_3 + 2H_2O \ \rightarrow H_2SO_4 + \ H_2O \qquad (R3)$$

The reaction (R1) is the rate-limiting step of this formation pathway (Finlayson-Pitts
and Pitts, 2000), so our proxy will consider the two major processes that determine the
abundance of gaseous sulfuric acid: the formation of gaseous sulfuric acid from
reactions between $SO_2$ and OH radicals, and the loss of gaseous sulfuric acid due to its
condensation onto pre-existing particles (Dal Maso et al., 2002; Kulmala et al., 2012;
Pirjola et al., 1999).

The rate of change of sulfuric acid concentration can be written as Eq. (3)

(Mikkonen et al., 2011):

$$d[H_2SO_4]/dt = k \ \cdot [OH] \ \cdot [SO_2] - [H_2SO_4] \ \cdot CS \qquad (3)$$

where $k$ is a temperature-dependent reaction constant (DeMore et al., 1997). To
simplify the calculation, the production and loss of sulfuric acid can be assumed to be
at pseudo steady-state (Mikkonen et al., 2011; Petäjä et al., 2009). Then the sulfuric
acid concentration can be written as Eq. (4).

$$[H_2SO_4] = k \ \cdot \ [OH] \ \cdot [SO_2] \ \cdot CS^{-1} \qquad (4)$$





Atmospheric OH radical measurements represent a major challenge as well. Since
previous studies suggest that the OH radical concentration is strongly correlated with
the intensity of UVB, [OH] could be replaced with UVB intensity in the proxy equation
(Petäjä et al., 2009; Rohrer and Berresheim, 2006). Although photolysis of $O_3$
$(\lambda < 320\,nm)$ and subsequent reactions with $H_2O$ are considered to be the dominant
source of OH radicals in the atmosphere (Logan et al., 1981), recent studies argue that
photolysis of HONO $(\lambda < 400\,nm)$ is a potentially important OH radical formation
pathway (Hendrick et al., 2014; Kleffmann, 2007; Su et al., 2011; Villena et al., 2011).
Thus, we attempt to introduce both $O_3$ and HONO into the proxy equation and evaluate
their effects on the concentration of OH radicals.
In practice, the values of the exponential factors in nonlinear fitting procedures are
rarely equal to 1 (Mikkonen et al., 2011), so we replaced the factors $x_i$ with $x_i^{w_i}$ in
the proxy, where $x_i$ can be an atmospheric variable and $w_i$ defines $x_i$' weight in the
proxy. Since $k$ is a temperature-dependent reaction constant and varies within a 10 %
range (in the atmosphere temperature range of 267.6 - 292.6 K), we further replaced $k$
with a scaling factor $k_0$ that is also used in the proxy methods built in Hyytiälä, Southern
Finland (Petäjä et al., 2009). As a result, the general proxy equation can be written as
Eq. (5), with the UVB intensity, $[SO_2]$, condensation sink (CS), $[O_3]$, and [HONO] (or
$[NO_x]$) as predictor variables:

$$[H_2SO_4] = f\left(k_0, x_i^{\omega_i}\right), \quad x_i = UVB, [SO_2], CS, [O_3], [HONO] \dots \quad (5)$$

The nonlinear curve-fitting procedures using iterative least square estimation for
the proxies of gaseous sulfuric acid concentration based on Eq. (4) were performed by
a MATLAB software.

**4 Results and discussion**
**4.1 General Characteristics of daytime sulfuric acid and atmospheric parameters**
Table 1 summarizes the mean, median and 5-95 % percentiles of gaseous sulfuric
acid concentrations and other variables measured during the daytime of the campaign.
The 5-95 % percentile ranges of the UVB intensity, $[SO_2]$, $[NO_2]$ and $[O_3]$ were 0-0.45
W m$^{-2}$, 0.9-11.4 ppbv, 3.3-61.4 ppbv and 3.5-23.3 ppbv, respectively. Compared with
the sites in the study by Mikkonen et al. (2011), Beijing was characterized with a factor
of 1.4-13.1 higher mean $[SO_2]$ but a factor of 3.4-5.4 lower mean $[O_3]$. The 5-95 %



252 percentile range of CS in Beijing was 0.01-0.24 s$^{-1}$, which is about 1-2 orders of

253 magnitude larger than corresponding value ranges in Europe and North America. The

254 concentration of gaseous sulfuric acid during this campaign was $(2.2 - 10.0) \times 10^6$

255 molecule cm$^{-3}$ was in a 5-95 % percentile range of, relatively similar to observed

256 elsewhere around the world. A diurnal mean concentration of 0.74 ppbv for HONO was

257 observed in this campaign, consistent with previous long-term HONO measurements

258 of about 0.48-1.8 ppbv (averaged values) in winter in Beijing (Hendrick et al., 2014;

259 Spataro et al., 2013; Wang et al., 2017), which is a factor of 4-10 higher than HONO

260 concentrations measured in Europe (Alicke et al., 2002, 2003). In addition, Beijing is

261 dry in winter with an ambient relative humidity generally lower than 60%.

263 **4.2 Correlations between [H$_2$SO$_4$] and atmospheric variables**

264  Table 2 summarizes the correlation coefficients between [H$_2$SO$_4$] and atmospheric

265 variables using a Spearman-type correlation analysis. Note that only correlations with

266 p-values smaller than 0.01 were included to ensure a statistical significance. Clearly,

267 the UVB intensity is an isolated variable that is independent of all the other variables

268 but that imposes a positive influence on O$_3$ because of photochemical formation of

269 ozone, and a negative influence on HONO because of HONO's photochemical

270 degradation. The sulfuric acid concentration shows positive correlations with all the

271 other variables. The correlation coefficients between [H$_2$SO$_4$] and [SO$_2$] and between

272 [H$_2$SO$_4$] and UVB intensity are 0.74 and 0.46, respectively, consistent with the accepted

273 formation pathway of gaseous sulfuric acid from the reaction between SO$_2$ and OH

274 radicals. Accordingly, [O$_3$] and [HONO] show positive correlations with [H$_2$SO$_4$]

275 because both O$_3$ and HONO could be precursors of OH radicals. Surprisingly, a high

276 positive correlation coefficient (0.6) was found between [H$_2$SO$_4$] and CS, which is in

277 contrast to the conventional thought that CS describes the loss of gaseous sulfuric acid

278 molecules onto pre-existing particles and thus should show a negative correlation. CS

279 correlates well with [SO$_2$] ($r = 0.83$) and [NO$_2$] ($r = 0.77$): a high CS value, as an

280 indicator of an atmospheric particle pollution, is thus usually accompanied with a high

281 concentration of both SO$_2$ and NO$_2$ in urban China, indicating co-emissions. A strong

282 correlation between [HONO] and [NO$_2$] ($r = 0.88$) in our measurement is supported by

283 the fact that HONO can be heterogeneously formed by reactions of NO$_2$ on various

284 surfaces (Calvert et al., 1994).

285  Since the UVB intensity and [SO$_2$] have been reported as the dominating factors



for the formation of sulfuric acid (Mikkonen et al., 2011; Petäjä et al., 2009), we further
explored the relationship of the measured sulfuric acid concentrations with the UVB
intensity and $[SO_2]$ using the nonlinear curve-fitting method with a single variable.
Figure 1a presents a scatter plot of $[H_2SO_4]$ against the UVB intensity, color-coded by
$[SO_2]$. A good correlation with a clear lamination by $[SO_2]$ is evident, indicating that
the UVB intensity and $[SO_2]$ together play an important role in the formation of sulfuric
acid. A similar scatter plot (Figure 1b) of $[H_2SO_4]$ against $[SO_2]$, color-coded by the
UVB intensity, leads to a similar conclusion.

**4.3 Proxy construction**
Similar to the non-linear proxies suggested by Mikkonen et al. (2011), we tested a
number of proxies for gaseous sulfuric acid, listed in Table 3 with their respective fitting
parameters and performance summarized in Table 4. The scatter plots of observed
$[H_2SO_4]$ *versus* predicted values given by proxies are presented in Fig. S1. In these
proxies, the concentration of a gaseous species is in the unit of molecule cm$^{-3}$, the unit
of the UVB intensity is W m$^{-2}$, the unit of CS is s$^{-1}$, and $k_0$ is a scaling factor.
The proxy N1 was built by using the UVB intensity and $[SO_2]$ as the source terms
and CS as the sink term, which follows the conventional idea of the $H_2SO_4$ formation
and loss in the atmosphere. CS was then removed from this proxy to examine the
performance of the proxy N2 that have the UVB intensity and $[SO_2]$ as the only
predictor variables. Since the formation of OH radicals in the atmosphere depends on
precursors in addition to UVB, we further attempted to introduce the OH precursor term
into the $H_2SO_4$ proxy. The proxies N3 and N4 were built by introducing $O_3$ as the only
OH precursor to evaluate its influence on the formation of sulfuric acid. Furthermore,
we added HONO as another potential precursor for OH radicals, resulting in the proxies
N5 and N6. Lastly, the proxy N7 was built by replacing [HONO] with $[NO_2]$ because
firstly, HONO is not regularly measured, and secondly, a good linear correlation
between [HONO] and $[NO_2]$ was generally observed in the daytime during this
campaign, although higher [HONO]/$[NO_2]$ ratios were observed in the morning due to
the accumulation of HONO during the night (Figure 2). RH was not considered in the
current study because the introduction of RH into the proxy did not yield significantly
better results in the Mikkonen et al. study (2011). In addition to the correlation
coefficient (R), Mean absolute error (MAE) was used to evaluate the performance of
proxies in the statistical analysis.



As shown in Table 4, the correlation coefficients are in the range of 0.83-0.86 and
MAEs are in the range of $(0.94 - 1.03) \times 10^6$ molecule cm$^{-3}$. The exponents for the
UVB intensity range from 0.13 to 0.16, and those for [SO$_2$] generally range from 0.38
to 0.41, except in case of the proxy N6 (b=0.33). The obtained exponent $b$ for [SO$_2$]
is significantly smaller than 1 unlike assumed in Eq. (3), mainly because [SO$_2$] is also
an indicator of air pollution that usually influences the sinks of both OH radicals and
sulfuric acid. The exponent for [SO$_2$] ranged from 0.5 to 1.04 in the previous proxy
study for European and North American sites (Mikkonen et al., 2011), including values
from 0.48 to 0.69 in Atlanta, GA, USA, which was probably quite a polluted site
because the measurements were conducted only 9 km away from a coal-fired power
plant. The obtained value range of the exponent $b$ for [SO$_2$] in our study is probably
related to the urban nature of Beijing. The value of exponent c for CS in the proxy N1
is as low as 0.03, which either might be due to the covariance of CS and certain H$_2$SO$_4$
sources that cancels the dependence on CS, or it might indicate that CS is actually
insufficient in regulating the H$_2$SO$_4$ concentration, as recently suggested by Kulmala et
al. (2017). By comparing the proxies N1 and N2, we can see that CS plays a minor role
because the exponents of [SO$_2$] and UVB, the overall correlation coefficient and the
MAE are almost identical with and without CS. We can see the negligible role of CS
also when comparing the results of the proxies N3 and N4 where O$_3$ is considered.
However, the role of CS becomes evident between the proxies N5 and N6 when HONO
is introduced: the exponents of [SO$_2$], [O$_3$], and [HONO] significantly increased when
taking into account the CS, suggesting that the covariance between HONO and CS can
explain, at least partially, the close-to-zero exponent of CS in the proxies N1-N4. In
addition, when [O$_3$] is introduced as the only precursor for OH radicals, minor
improvements in the correlation coefficient and MAE were obtained, as suggested by
comparing the proxies N3 and N1. When both [O$_3$] and [HONO] were introduced as
OH precursors in the proxies N5-N7, MAE and correlation coefficient significantly
improved. Altogether, these observations suggest that it is crucial to introduce HONO
into the proxy, both in our study and also likely for the previous work where the
exponent of CS is close-to-zero (Mikkonen et al., 2011).
Although so far the proxy N5 had the best fitting quality, it is impractical to
explicitly include [HONO] because HONO measurements are very challenging. As
shown in Fig. 2, [HONO] and [NO$_2$] are tended to correlate linearly with each other in
the daytime during this campaign, with a linearly fitted [HONO]/[NO$_2$] ratio of around





0.03 and a mean absolute error (MAE) of 0.3 ppbv. Similar, strong linearity was
observed in a previous study by Hao et al. (2006) who attributed this observation to the
heterogeneous conversion of $NO_2$ to HONO. Only occasionally slightly higher
[HONO]/[$NO_2$] ratios in the morning could be seen, which might be due to the
deviation from the steady state. Bernard et al. (2016) reported that [$NO_2$] has a similar
diurnal behavior to that of [HONO] and hence the ratio of [HONO]/[$NO_2$] varies
slightly during the diurnal cycle. Therefore, due to the good correlation, the proxy N7
replaces [HONO] by [$NO_2$], a more easily measured variable, and performs equally
well with the proxy N5.
Clearly, the proxy N2 provides the simplest parameterization, but the proxies N5
and N7 result in the best fitting quality because of the introduction of [HONO]. Figure
3a and 3b present the averaged and relative deviation of calculated sulfuric acid
concentrations according to the proxies N2 and N7, respectively, as a function of linear
bins of measured sulfuric acid concentrations. The averaged and relative deviation are
defined by Eq. (6) and Eq. (7), respectively, assuming that there are a number of $j$ data
points, both measured and calculated, in the $i^{th}$ bin.

$$\textit{Averaged deviation} = \frac{1}{n} \cdot \sum_{j=1}^{n} ([H_2SO_4]_{proxy,ij} - [H_2SO_4]_{meas.,ij}) \quad (6)$$

$$\textit{Relative deviation} = \frac{1}{n} \cdot \sum_{j=1}^{n} \frac{([H_2SO_4]_{proxy,ij} - [H_2SO_4]_{meas.,ij})}{[H_2SO_4]_{meas.,ij}} \quad (7)$$


The performance of the proxy N7 is considerably better than that of the proxy N2
in the sulfuric acid concentration range of $(2.0 - 15) \times 10^6$ molecule cm$^{-3}$, which
covers most measured concentrations of sulfuric acid. The relative deviation is less than
24% for all the bins in case of the proxy N7, rising up to 29% in case of the proxy N2.
For the first bin in a range of $(1.0 - 2.0) \times 10^6$ molecules cm$^{-3}$, both proxy N2 and
N7 show small averaged deviations but the biggest relative deviations, which is due to
the smallest denominators. Since we want to make it clear in which bins the predicted
values are overestimated or underestimated in Fig. 3a and 3b, the calculation method
from Eq. (6) and Eq. (7) would make the positive derivations and negative derivations
counteracted to some extent, which is obvious in the middle bins.





**4.4 Comparison of measured and predicted [H₂SO₄]**

A comparison between measured and predicted [$H_2SO_4$] was performed. Figure 4 includes calculated results from the proxies N2 and N7 as well as from a proxy constructed according to measurement in a boreal forest site, Finland, *i.e.*, Eq (1) (Petäjä et al., 2009). The measured daytime [$H_2SO_4$] on 10 March, 2018, was above $4 \times 10^6$ molecules cm$^{-3}$ with a time resolution of 5 min. The predicted [$H_2SO_4$] using the proxies N2 and N7 both track the measured [$H_2SO_4$] pretty well, even when an unexpected dip in the sulfuric acid concentration was observed at around 10:00-11:00. The performance of the proxy N7 is better than that of proxy N2 during the entire day, consistent with our results in Fig. 3. The proxy by Petäjä et al. (2009) underestimated the concentrations of sulfuric acid by a factor of 20 or so, which can be attributed to the very different values of CS between Beijing and the boreal forest. The fact that [$H_2SO_4$]$_{Petäjä\ et\ al.}$ does not track the measured [$H_2SO_4$] even after including a scaling factor indicates that proxies are site-specific and do not necessarily work well in locations other than where they have originally been developed for. In addition, the direct performance comparison between the proxy N2 and the proxy by Petäjä et al. (2009) indicates the importance of assigning exponential weights to variables in the nonlinear fitting procedures, which is consistent with results by Mikkonen et al. (2011).

**5 Summary and conclusions**

Sulfuric acid is a key precursor for atmospheric new particle formation. In this study, we constructed a number of proxies for gaseous sulfuric acid concentration according to our measurements in urban Beijing during the winter. According to the obtained proxies and their performance, the UVB intensity and [$SO_2$] were the dominant influencing factors. Hence, the simplest proxy (Proxy N2) only involves UVB intensity and [$SO_2$] as shown by Eq. (8).

$$[\boldsymbol{H_2SO_4}] = \boldsymbol{280.05} \cdot \boldsymbol{UVB^{0.14}} \cdot [\boldsymbol{SO_2}]^{\boldsymbol{0.40}} \qquad (8)$$

This proxy resulted in a relative deviation of up to 29 %.

For the best proxy accuracy, [$O_3$] and [HONO] as well as CS should be included (Proxy N5), as shown by Eq. (9):





$$[H_2SO_4] = 0.0072 \cdot UVB^{0.15} \cdot [SO_2]^{0.41} \cdot CS^{-0.17} \cdot ([O_3]^{0.36}$$
$$+ [HONO]^{0.38}) \quad (9)$$

Since HONO measurements are not a regular practice, we can further replace [HONO]
with [NO$_2$], shown in Eq. (10), which can be justified by the strong linear correlation
between [HONO] and [NO$_2$] observed in this study:

$$[H_2SO_4] = 0.0013 \cdot UVB^{0.13} \cdot [SO_2]^{0.40} \cdot CS^{-0.17} \cdot ([O_3]^{0.44}$$
$$+ [NO_2]^{0.41}) \quad (10)$$

We consider this last proxy more reasonable than the others due to the following reasons:
first, it makes the equation physically meaningful as the CS starts to be involved as a
sink term, and second, the absolute and relative fitting error were reduced considerably
compared with the other proxies. Overall, this suggests that the photolysis of O$_3$ and
HONO are both important OH sources in urban Beijing.
As a summary, we recommend using the simplest proxy (proxy N2) and a more
accurate proxy (Proxy N7) for calculating daytime gaseous sulfuric acid concentrations
in the urban Beijing atmosphere. It is clear that the current proxies are based on only a
month-long campaign of sulfuric acid measurements in urban Beijing during winter.
Given the dramatic reduction in the concentration of SO$_2$ in recent years (Wang et al.,
2018) and the strong dependence of calculated [H$_2$SO$_4$] on [SO$_2$], the performance of
the proxies in the past and future years remain to be evaluated. Nevertheless, our work
here shows the importance of heterogeneous chemistry as a potential source of OH
radicals in an urban air; however, the proxies might be site-specific and should be
further tested before their application to other Chinese megacities.

**Author contributions**
LW designed this study. YL (Yiqun Lu), CY, YF, YC, YL (Yiliang Liu), GY, YW, YZ, RY, RB
and CD conducted the field campaign. YL (Yiqun Lu) analyzed data with contributions from
LW and all the other co-authors. YL (Yiqun Lu) and LW wrote the manuscript with
contributions from all the other co-authors.

**Acknowledgement**
This study was financially supported by the National Key R&D Program of China



(2017YFC0209505), and the National Natural Science Foundation of China (41575113,

91644213).





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



Table 1 Mean, median, 5-95 % percentiles of key atmospheric variables and [H₂SO₄] in the daytime.

| | UVB (W m⁻²) | [SO₂] (ppbv) | CS (s⁻¹) | [O₃] (ppbv) | [HONO] (ppbv) | [NO₂] (ppbv) | [H₂SO₄] (× 10⁶ molecule cm⁻³) | RH (%) |
|---|---|---|---|---|---|---|---|---|
| mean | 0.17 | 4.6 | 0.11 | 10.5 | 0.74 | 25.3 | 5.4 | 28 |
| median | 0.14 | 3.7 | 0.11 | 9.0 | 0.51 | 23.0 | 4.9 | 26 |
| 5-95% pencentiles | 0.00-0.45 | 0.9-11.4 | 0.01-0.24 | 3.5-23.3 | 0.09-2.65 | 3.3-61.4 | 2.2-10.0 | 9-59 |





Table 2 Correlation coefficients (Spearman type) between [H$_2$SO$_4$] and atmospheric variables in the daytime. Only correlation coefficients with p-values less than 0.01 are included to ensure a statistical significance.

|  | UVB | [SO$_2$] | CS | [O$_3$] | [HONO] | [NO$_2$] | [H$_2$SO$_4$] |
|---|---|---|---|---|---|---|---|
| UVB | 1 | / | / | 0.14 | -0.23 | / | 0.46 |
| [SO$_2$] |  | 1 | 0.83 | 0.25 | 0.64 | 0.70 | 0.74 |
| CS |  |  | 1 | 0.36 | 0.75 | 0.77 | 0.60 |
| [O$_3$] |  |  |  | 1 | / | / | 0.29 |
| [HONO] |  |  |  |  | 1 | 0.88 | 0.39 |
| [NO$_2$] |  |  |  |  |  | 1 | 0.53 |
| [H$_2$SO$_4$] |  |  |  |  |  |  | 1 |


Table 3 Proxy functions for the nonlinear fitting procedure.

| Proxy | Equation[#] |
|---|---|
| N1 | $k_0 \cdot UVB^a \cdot [SO_2]^b \cdot CS^c$ |
| N2 | $k_0 \cdot UVB^a \cdot [SO_2]^b$ |
| N3 | $k_0 \cdot UVB^a \cdot [SO_2]^b \cdot CS^c \cdot [O_3]^d$ |
| N4 | $k_0 \cdot UVB^a \cdot [SO_2]^b \cdot [O_3]^d$ |
| N5 | $k_0 \cdot UVB^a \cdot [SO_2]^b \cdot CS^c \cdot ([O_3]^d + [HONO]^e)$ |
| N6 | $k_0 \cdot UVB^a \cdot [SO_2]^b \cdot ([O_3]^d + [HONO]^e)$ |
| N7 | $k_0 \cdot UVB^a \cdot [SO_2]^b \cdot CS^c \cdot ([O_3]^d + [NO_2]^f)$ |

[#]UVB is the intensity of ultraviolet radiation b in W cm$^{-3}$; [SO$_2$] is the concentration of sulfur dioxide in molecule cm$^{-3}$; CS is the condensation sink in s$^{-1}$; [O$_3$] is the concentration of ozone in molecule cm$^{-3}$; [HONO] is the concentration of nitrous acid in molecule cm$^{-3}$; [NO$_2$] is the concentration of nitrogen dioxide in molecule cm$^{-3}$; $k_0$ is a scaling factor.





Table 4 Results of the nonlinear fitting procedure for different proxy functions, together with correlation coefficient (R, Pearson type) and mean absolute error (MAE).

| Proxy | $k_0$ | $a$ | $b$ | $c$ | $d$ | $e$ | $f$ | $R$ | $MAE$ ($\times 10^6$ molecule $cm^{-3}$) |
|---|---|---|---|---|---|---|---|---|---|
| N1 | 515.74 | 0.14 | 0.38 | 0.03 | | | | 0.83 | 1.03 |
| N2 | 280.05 | 0.14 | 0.40 | | | | | 0.83 | 1.03 |
| N3 | 9.95 | 0.13 | 0.39 | -0.01 | 0.14 | | | 0.85 | 1.00 |
| N4 | 14.38 | 0.13 | 0.38 | | 0.14 | | | 0.85 | 1.00 |
| N5 | 0.0072 | 0.15 | 0.41 | -0.17 | 0.36 | 0.38 | | 0.86 | 0.94 |
| N6 | 2.38 | 0.14 | 0.33 | | 0.24 | 0.24 | | 0.85 | 0.98 |
| N7 | 0.0013 | 0.13 | 0.40 | -0.17 | 0.44 | | 0.41 | 0.86 | 0.95 |



**Figure Captions**

**Figure 1.** Correlations (a) between [H₂SO₄] and UVB intensity, and (b) between [H₂SO₄] and [SO₂]. $k_m$ is a constant term.

**Figure 2.** Correlation between [HONO] and [NO₂]. The black line represents a linear fitting with a zero intercept.

**Figure 3.** Performance assessments of proxy N2 and proxy N7. The averaged deviation and the relative deviation in the plots are defined by Eq. (6) and Eq. (7) and used to evaluate the performance of proxy N2 and N7, respectively. "Overlap" refers to the smaller values between proxy N2 and proxy N7, and the larger ones are indicated by the color code of proxies N2 and N7.

**Figure 4.** Comparison of measured [H₂SO₄], $[H_2SO_4]_{N2}$, $[H_2SO_4]_{N7}$ and $[H_2SO_4]_{\text{Petäjä }et\ al.}$ on 10 March, 2018 with a time resolution of 5 min.



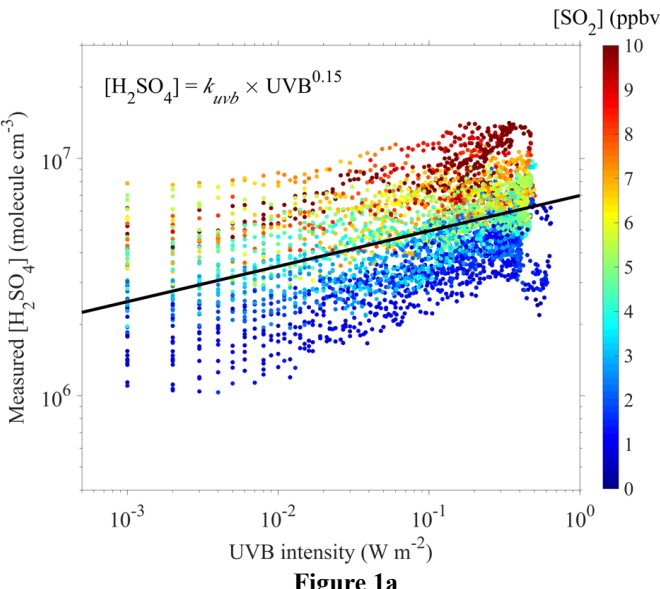

**Figure 1a**

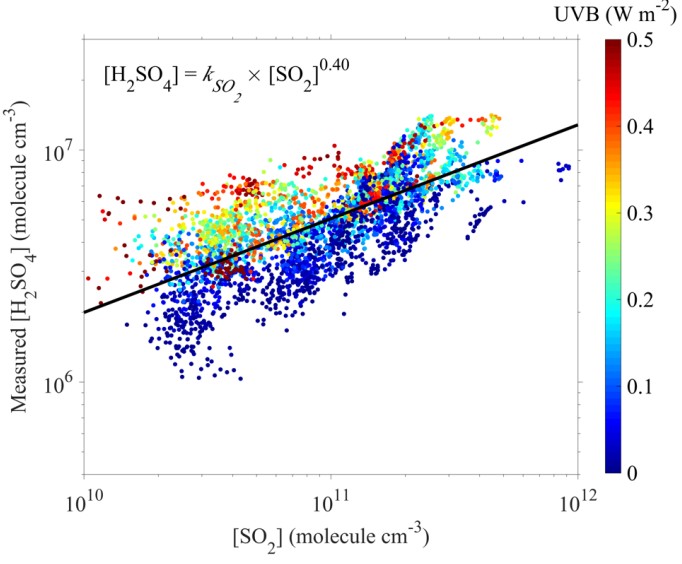

**Figure 1b**



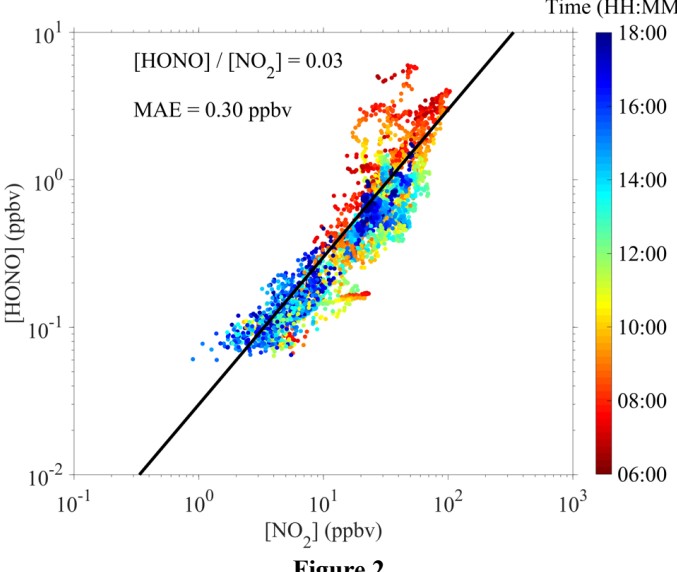

**Figure 2**



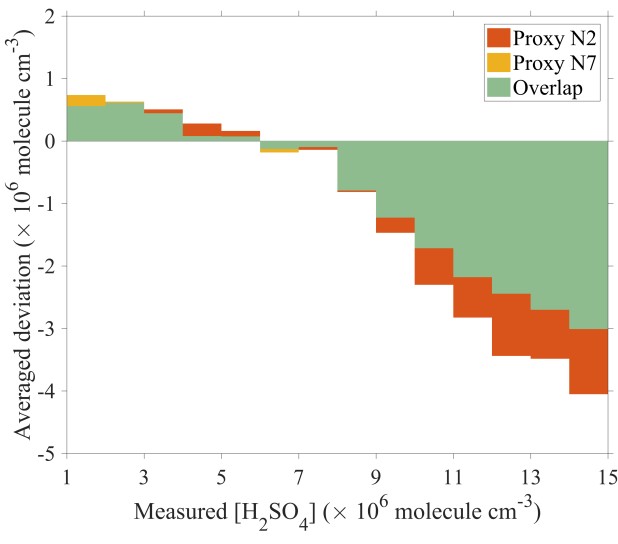

**Figure 3a**

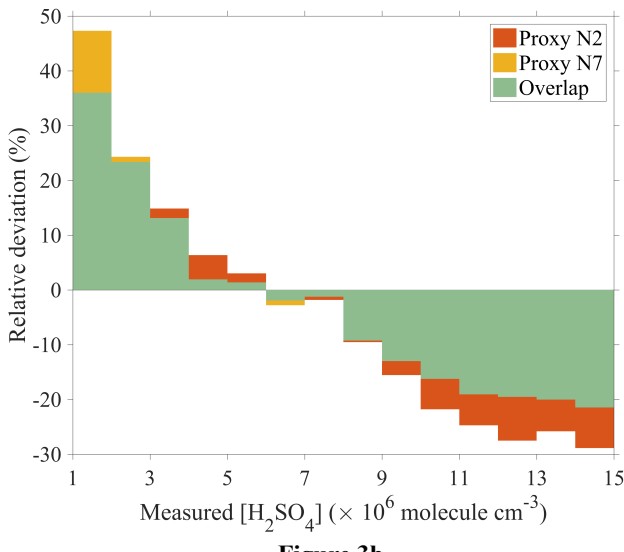

**Figure 3b**



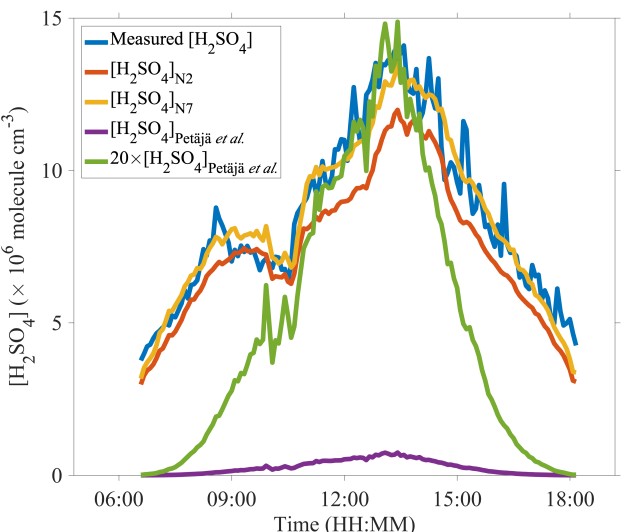

**Figure 4**