# Peer review of "A proxy for atmospheric daytime gaseous sulfuric acid concentration in urban Beijing"

_Atmospheric Chemistry and Physics, 2018_

## Referee Comment (RC1) · Anonymous Referee #2 · 19 Dec 2018

The manuscript by Lu et al. evaluates different equations for the calculation of gas-phase sulfuric acid from proxy parameters (like SO2, UVB radiation, condensation sink, etc.). Different sets of parameters are tested and the performance of the proxy equations is evaluated against in-situ measurements of gas phase H2SO4 (measured with a chemical ionization mass spectrometer) and the measured proxy parameters. The measurements took place in Beijing during February/March 2018. While similar proxy expressions for sulfuric acid have been derived for other locations (see Mikkonen et al., 2011, ACP, 11(21), 11319–11334) this is the first one applying such methods for the conditions of a Chinese mega city. Unlike previous studies, the present one includes also HONO as an OH precursor and not just ozone. This leads to a slightly better correlation coefficient between the proxy-derived and measured H2SO4. The

manuscript is scientifically relevant and should be published in ACP after the authors have considered the points listed below. Besides the general comments, I have also listed a number of suggestions to improve the language.

Comments:

(1) Page 1, line 25: "remains a major challenge" is a bit exaggerated, please revise this statement

(2) Page 2, line 43: How is it known that HONO is formed heterogeneously? Isn't the gas-phase reaction between OH and NO also an efficient source?

(3) Page 6, line 169: Is the upper size limit of 700 nm sufficient to include all relevant particles contributing to the condensation sink? During dust events, larger particles can probably contribute quite significantly to the CS.

(4) Page 8, line 231 to 234: It should be explained in more detail how k is calculated and how it relates to k0.

(5) Page 9, line 281 to 284: Could the correlation between HONO and NO2 also be caused by the gas phase reaction between OH and NO (because NO correlates probably strongly with NO2)?

(6) Page 10, line 317 to 319: The mean absolute error does not seem to be the best metric for evaluating the fitting of data that vary over more than one order of magnitude. This can lead to a bias where the high values are well represented but the relative deviation for the small values can be large. A better metric could be the average ratio (sum over all max(Vi_measured , Vi_fitted) / min(Vi_measured , Vi_fitted), where Vi is a data point for the considered parameter, e.g. [H2SO4]).

(7) Page 11, line 346/347: The improvement relates to an increase of R from 0.85 to 0.86; this does not justify the word "significantly"; "improved slightly" instead of "improved significantly" is more appropriate.

[Figure]

(8) Page 13, line 396: Aren't the different values of CS taken into account in the calculation, or do the authors mean that the different exponents for CS make the difference? What is the exponent for the CS in the Petäjä et al. study?

(9) Page 14, line 439: How can the proxies be evaluated for past measurements? If measurements exist for both sulfuric acid and the proxy parameters, these should be included in the present study.

(10) Page 14, line 440: The importance of heterogeneous chemistry cannot be concluded from the presented data; this statement should be removed or supported with further data.

(11) Section 5: Discussion on the expected seasonality of the proxy-H2SO4 relation should be discussed. Very likely, the exponents can have different values for other meteorological condition, e.g., when desert dust contributes to the particle burden. In such a case, the exponent for the CS could become negative. This possibility should be mentioned/discussed.

(12) Table 1: The values for ozone are quite low. Are these low values typical for wintertime conditions in Beijing?

(13) Table 3: What is the definition of the scaling factor k0? What are its units?

(14) Figure 3: The yellow areas are hardly visible; it would probably be better to use colored lines instead of filled areas for this figure. In addition, the shape of the curves suggests a pronounced bias (the high values are on average underestimated, while the low values are overestimated). This bias can also be seen in the SI Figures. Using a different metric for the fitting (ratios instead of absolute differences, see above) could solve this issue.

Language:

Page 2, line 46: "less than" rather than "up to"?

Page 3, line 57: "involve" instead of "demand"

Page 3, line 73/74: delete "using a passive CIMS"

Page 3, line 76: delete "associated equilibrium or"

Page 3, line 79: replace "will be" with "are"

Page 4, line 90: replace "supposing" with "assuming"

Page 4, line 102: delete ", a proxy for condensational sink"

Page 4, line 108: delete "that characterized"

Page 4, line 114: replace "between" with "in"

Page 6, line 156: "the sample flow"

Page 6, line 177: "dependent on" instead of "that could be defined as a function of"

Page 8, line 230: What is meant by the symbol xi'?

Page 9, line 255: "in the 5-95% percentile range" instead of "was in . . . range of"

Page 9, line 280: delete "an"

Page 10, line 290: "lamination" does not seem to be the right word here, maybe better to use "layering with"

Page 10, line 305: "has" instead of "have"

Page 11, line 352: either delete "are tended to" or use "tend to"

Page 12, line 382: Do the authors mean "deviations" instead of "derivations"?

Page 13, line 390: "when averaged to a time resolution of 5 min" instead of "with a time resolution of 5 min"

Page 13, line 411: Please provide the units for the parameters in this equation.

[Figure]

Page 14, line 418/419: Please provide the units for the parameters in this equation.

Page 14, line 425/426 Please provide the units for the parameters in this equation.

Page 14, 433/434: Please provide also the equation numbers and not just the proxy numbers (N2 and N7).

---

## Referee Comment (RC2) · Anonymous Referee #1 · 20 Dec 2018

This study examines the relationship of [SO2] to [H2SO4] as a function of light intensity, particle concentration, and other gas phase reactants in Beijing. As the authors point out, this relationship is likely different in Beijing than in European and US cities. Overall, this study is straight forward and generally useful for research conducted in megacities. However, several issues should be address before this study can be considered for publication in ACP.

Major comments: P5 147: Sulfuric acid concentration was measured using nitrate LToF-CIMS. It would be useful for the reader to know more details on how the sulfuric acid concentration was determined from the signals of the instrument. Does this measurement include sulfuric acid in molecular clusters (i.e. is fragmentation contributing to the sulfuric acid signal?) What are the estimated uncertainties of the sulfuric acid

measurement? How do these uncertainties compare to the model predicted amounts?

P6 162: Along these same lines, the authors comment that the calibration coefficient takes into account diffusion losses in the sampling line. Was this loss measured? It is a bit surprising that the calibration coefficient that Kurten et al. (2012) determined was $1.1 \times 10^{10}$ cm-3 is the same in this study. I would have thought differences in instrument and sampling line losses (1.6 m is quite long) would have impacted this number. The authors should more clearly lay out how the sulfuric acid concentration was determined since it is an integral measurement for this paper.

P6 L 176: The Fuchs-Sutugin transition kernel is used. There is associated error with using this kernel in the transition regime where sulfuric acid condenses on pre-existing particles. Can the authors comment on this error? How sensitive is the fitting parameters to changes in the collision kernel? It may be helpful for the authors to use the empirically-derived collision kernel for the full regime from (Gopalakrishnan and Hogan Jr., 2011) to help limit the uncertainties from this parameter.

P5 134: The authors state that two months of measurements were conducted. It is not clear from the paper if all two months of measurements were used to determine the proxy relationship. Have the authors examined how the proxy relationship changes from day to day? Or week to week? The authors correctly state that the proxy relationship likely depends on location but does it also depend on time? It is possible that other processes that affect sulfuric acid concentrations (like Criegee intermediates) are not captured in the proxy relationship may play a larger role during some times of day than others.

P13 line 389: If two months of measurements were taken, why was only one day used to compare measured to predicted sulfuric acid concentrations? How does the comparison for the other days look? It's not necessary to add graphs of these comparison, but a few lines stating the comparison for other days is necessary for the reader to determine how useful this proxy relationship is.

P13 line 396: Authors state that the proxy relationship developed for the boreal forest and applied to Beijing is a factor of 20 too low due to differences in CS. Why didn't the authors use the Beijing CS values when applying Petäjä et al.'s proxy relationship? Would the differences between measured and predicted from Petäjä then be smaller?

Figure 4: It would be useful for the reader to see timelines of all the measured concentrations that go into the proxy relationships as well.

Minor comments:

P1 Line 28: desirable for the atmospheric. . .

P1 36-27 change one of the "using"

P3 Line 57: sulfuric acid DMA system. The citation for Petäjä et al. (2011) might not be the best. Several studies have pointed out potential experimental issues with this study (Jen et al., 2014; Kürten et al., 2014).

P3 line 57: demand participation is a strange phrase. Maybe necessitates participations?

P3 line 59: Would be worth reading and citing (Kuang et al., 2012) for sulfuric acid growth rates.

P3 line 62: Knowing sulfuric acid concentrations prior to a nucleation event is also important.

P3 Line 68: NO3- and ligands.

P3 line 68: CIMS is actually a pretty broad class of instruments. The low detection limit for sulfuric acid is because the instrument ionizes and samples at atmospheric pressure, which is different than the traditional CIMS.

P3 line 80: (Chen et al., 2012) shows a nice figure of sulfuric acid concentrations measured at numerous locations around the world. Not critical to add the citation but

worth taking a look at.

P3 line 83: has been proven

P4 line 105: After reading this, the reader will naturally wonder why is there a positive correlation between CS and sulfuric acid concentration?

P4 line 108: locations that characterize with an... one or two of those words are not correct.

P4 line 110: Please state the range of CS in addition to how much higher it is compared to other locations.

P4 line 113: For north America: how do these numbers compare to Mexico City?

P5 119: OH radicals

P5 119: remove the not only and but also. It is harder to read with them there.

P5 line 128: Criegee should be capitalized

P6 line 153: was guided through... strange phrasing

P6 line 154: Is this a custom-built inlet? If so, could the authors provide a diagram and write in the dimensions?

P6 line 160: CIMS was calibrated. How? It would be useful to describe this procedure in brief.

P6 line 164: should it be ToFTools?

P6 line 166: 1 nm. Is this mobility diameter?

P7 line 213: Authors should better justify pseudo-steady state assumption

P 8 paragraph starting on line 228: This was a difficult paragraph to understand. Can the authors better phrase it to explain the differences in parameters?

P 8 line 242: a matlab software. A custom-made one? Or just a function in matlab?

P 9: 1-2 orders of magnitude. Maybe change to 10-100 times higher to be more clear.

P9 line 261: 60% RH does not seem dry.

P9 272: I do not understand how the correlation coefficient numbers are consistent with accepted formation pathways? Does the formation pathways have powers that are less than 1?

P9 line 276: Authors should explain potential reasons why sulfuric acid positively correlates with CS.

P10 300: molecules cm-3 is normally written as just cm-3.

P10 line 316: Authors mention that proxy relationship is location specific. Why then did the authors use the justification for not including RH based upon conclusions drawn from a different location?

P11 line 324: "unlike assumed in Eq. (3)" wording seems incorrect

P11 line 324: The naming convention between the equations in table 3 and the equations in the paper is confusing. Which equation 3 does this line refer to?

Page 12 line 356: "Only occasionally slightly higher" too many adverbs. Rephrase

Page 12 line 356: The authors refer to a previous study to justify linearity of NO2 and HONO. Where was the location of this study? This paragraph is general is difficult to discern results from previous studies and results from this study. Please make this more clear.

Page 12 line 376: authors should specific that this cover sulfuric acid concentrations for this location. 10ˆ6 cm-3 does not cover sulfuric acid concentrations around the world.

Page 13 line 416: It is a bit confusing that the authors mention that proxy N5 is the most accurate when they spend most of the paper justifying the use of N7. Maybe change

the wording "for the best proxy accuracy" or consider rewording this section to make it a bit less confusing/

Page 14 line 439: I do not understand how this work has shown the importance of heterogenous chemistry as a potential source of OH. Was this mentioned somewhere else in the main paper? The authors should better justify this point if they want to include in the summary.

Figure 1-2: What day were these measurements done?

Figure 2: Can the authors explain why there seems to be clear break up group of points during the early morning that do not follow the linear trend?

Figure 4: As mentioned above, it would be useful to show the time lines for the other measured concentrations (CS, OH, NO2, etc. ) that the proxy model uses.

Works cited for this review: Chen, M., Titcombe, M., Jiang, J., Jen, C., Kuang, C., Fischer, M. L., Eisele, F. L., Siepmann, J. I., Hanson, D. R., Zhao, J. and Mc-Murry, P. H.: Acid–base chemical reaction model for nucleation rates in the polluted atmospheric boundary layer, Proc. Natl. Acad. Sci., 109, 18713–18718, doi:10.1073/pnas.1210285109, 2012. Gopalakrishnan, R. and Hogan Jr., C. J.: Determination of the Transition Regime Collision Kernel from Mean First Passage Times, Aerosol Sci. Technol., 45(12), 1499–1509, doi:10.1080/02786826.2011.601775, 2011. Jen, C. N., McMurry, P. H. and Hanson, D. R.: Stabilization of sulfuric acid dimers by ammonia, methylamine, dimethylamine, and trimethylamine, J. Geophys. Res. Atmospheres, 119, 2014JD021592, doi:10.1002/2014JD021592, 2014. Kuang, C., Chen, M., Zhao, J., Smith, J., McMurry, P. H. and Wang, J.: Size and time-resolved growth rate measurements of 1 to 5 nm freshly formed atmospheric nuclei, Atmos Chem Phys, 12, 3573–3589, doi:10.5194/acp-12-3573-2012, 2012. Kürten, A., Jokinen, T., Simon, M., Sipilä, M., Sarnela, N., Junninen, H., Adamov, A., Almeida, J., Amorim, A., Bianchi, F., Breitenlechner, M., Dommen, J., Donahue, N. M., Duplissy, J., Ehrhart, S., Flagan, R. C., Franchin, A., Hakala, J., Hansel, A., Heinritzi, M., Hutterli, M., Kangasluoma,

J., Kirkby, J., Laaksonen, A., Lehtipalo, K., Leiminger, M., Makhmutov, V., Mathot, S., Onnela, A., Petäjä, T., Praplan, A. P., Riccobono, F., Rissanen, M. P., Rondo, L., Schobesberger, S., Seinfeld, J. H., Steiner, G., Tomé, A., Tröstl, J., Winkler, P. M., Williamson, C., Wimmer, D., Ye, P., Baltensperger, U., Carslaw, K. S., Kulmala, M., Worsnop, D. R. and Curtius, J.: Neutral molecular cluster formation of sulfuric acid–dimethylamine observed in real time under atmospheric conditions, Proc. Natl. Acad. Sci., doi:10.1073/pnas.1404853111, 2014.

---

## Short Comment (SC1) · 9 Jan 2019

It is interesting to see how the sulphuric acid concentration can be approximated in highly polluted environment, as we did not have such data when we were making our paper Mikkonen et al. (2011). Even more interesting is, that your recommended proxy N2 is quite close to our second recommendation, simple proxy L3, having SO2 power to 0.5 when you have power of 0.4. In addition, I was surprised that the H2SO4 concentration was not higher than shown in Table 1. We had similar average concentrations in San Pietro Capofiume and considerably higher in Atlanta, even though they are less polluted environments. Could you add a comment on that?

I just want to ask about Figure 4: Why only one day, and not averages over whole period

such that uncertainty would also be indicated, is shown in the figure? In addition, why comparison only to Boreal forest-proxy from Petäjä et al, why not to Mikkonen et al., who had data from multiple sites?

A minor comment on the use of p-value as a screening factor for correlation (in line 266): it is not recommended. See e.g. Greenland et al. (2016): DOI 10.1007/s10654-016-0149-3

---

## Author Comment (AC1) · 30 Jan 2019

**RE: A point-to-point response to reviewers' comments**

"A proxy for atmospheric daytime gaseous sulfuric acid concentration in urban Beijing" (acp-2018-1132) by Yiqun Lu, Chao Yan, Yueyun Fu, Yan Chen, Yiliang Liu, Gan Yang, Yuwei Wang, Federico Bianchi, Biwu Chu, Ying Zhou, Rujing Yin, Rima Baalbaki, Olga Garmash, Chenjuan Deng, Weigang Wang, Yongchun Liu, Tuukka Petäjä, Veli-Matti Kerminen, Jingkun Jiang, Markku Kulmala, Lin Wang

We are grateful to the helpful comments from this anonymous referee, and have carefully revised our manuscript accordingly. A point-to-point response to the comments, which are repeated in italic, is given below.

In addition to the reviewers' comments, we have noticed and corrected a key typo from our previous version of manuscript. "The [$NO_2$] concentration" in our manuscript is in fact "the [$NO_x$] concentration". Correction of this term does not lead to changes in our conclusions.

**Reviewer #2's comments:**

*The manuscript by Lu et al. evaluates different equations for the calculation of gas-phase sulfuric acid from proxy parameters (like $SO_2$, UVB radiation, condensation sink, etc.). Different sets of parameters are tested and the performance of the proxy equations is evaluated against in-situ measurements of gas phase $H_2SO_4$ (measured with a chemical ionization mass spectrometer) and the measured proxy parameters. The measurements took place in Beijing during February/March 2018. While similar proxy expressions for sulfuric acid have been derived for other locations (see Mikkonen et al., 2011, ACP, 11(21), 11319–11334) this is the first one applying such methods for the conditions of a Chinese mega city. Unlike previous studies, the present one includes also HONO as an OH precursor and not just ozone. This leads to a slightly better correlation coefficient between the proxy-derived and measured $H_2SO_4$. The manuscript is scientifically relevant and should be published in ACP after the authors have considered the points listed below. Besides the general comments, I have also listed a number of suggestions to improve the language.*

Reply: We are very grateful to the positive viewing of our manuscript by Reviewer #2, and have now revised our manuscript accordingly.

**General comments:**

1. *Page 1, line 25: "remains a major challenge" is a bit exaggerated, please revise this statement*

Reply: This statement now reads (L25) "but its measurement remains a difficulty".

2. *Page 2, line 43: How is it known that HONO is formed heterogeneously? Isn't the gas-phase reaction between OH and NO also an efficient source?*

Reply: We agree with reviewer #2 that HONO can be formed from both homogeneous and heterogeneous processes. "heterogeneously-formed" has been removed.

3. *Page 6, line 169: Is the upper size limit of 700 nm sufficient to include all relevant particles contributing to the condensation sink? During dust events, larger particles can probably contribute quite significantly to the CS.*

Reply: The CS values were actually calculated based on the particle size distributions up to 10 μm. We have corrected our description of the upper size limit of the PSD system, which reads (L202) "…a conventional particle size distribution system (PSD, ~3 nm - 10 μm)".

On the other hand, particle number size distributions up to 700 nm will allow a reasonable calculation of CS, given the fact that most particles are smaller than 700 nm and that there were not significant dust events during our measurements.

4. *Page 8, line 231 to 234: It should be explained in more detail how k is calculated and how it relates to $k_0$.*

Reply: An explanation on how *k* is calculated has been added, which reads (L247-L255)

"…where *k* is a temperature-dependent reaction constant given by Eq. (5) (DeMore *et al.*, 1997; Mikkonen *et al.*, 2011).

$$k = \frac{A \cdot k_3}{(A + k_3)} \cdot exp\left\{k_5 \cdot \left[1 + log_{10}\left(\frac{A}{k_3}\right)^2\right]^{-1}\right\} \quad cm^3(molecule \cdot s)^{-1} \quad (5)$$

where $A = k_1 \cdot [M] \cdot (\frac{300}{T})^{k_2}$, $[M]$ represents the density of the air in molecule cm$^{-3}$ as calculated by $0.101 \cdot (1.381 \cdot 10^{-23} \cdot T)^{-1}$, $k_1 = 4 \cdot 10^{-31}$, $k_2 = 3.3$, $k_3 = 2 \cdot 10^{-12}$ and $k_5 = -0.8$."

We have also elaborated the explanation on how *k* relates to $k_0$, which reads (L316-L320) "Since *k* is a temperature-dependent reaction constant and varies within a 10 % range in the atmosphere temperature range of 267.6 - 292.6 K, *i.e.*, the actual atmospheric temperature variation in this study, we approximately regard *k* as a constant and use a new scaling factor $k_0$. This methodology has been used previously in the proxies of gaseous sulfuric acid in Hyytiälä, Southern Finland (Petäjä *et al.*, 2009)".

5.  *Page 9, line 281 to 284: Could the correlation between HONO and NO₂ also be caused by the gas phase reaction between OH and NO (because NO correlates probably strongly with NO₂)?*

Reply: As stated in the very beginning, "The [NO₂] concentration" in our previous manuscript is in fact "the [NOₓ] concentration" due to a key typo. Hence, the correlation coefficients (Spearman type) between NOₓ and HONO, between NO and HONO, and between NO₂ and HONO are 0.88, 0.74 and 0.88, respectively. Although a slightly better correlation between NO₂ and HONO was observed, we cannot exclude the role of the gas phase reaction between OH and NO, and the interconversion between NO and NO₂. In fact, we agree with reviewer #2 that HONO can be both homogeneously and heterogeneously formed, although heterogeneous formation from NO₂ is likely the reason for the daytime HONO production in urban Beijing (Liu *et al.*, 2014). In the revised manuscript, we have stated (L335-L339) that "A strong correlation between [HONO] and [NOₓ] (r = 0.88) in our measurement is supported by the fact that HONO can be either heterogeneously formed by reactions of NO₂ on various surfaces (Calvert et al., 1994) or homogeneously formed by the gas phase NO + OH reaction, between which the former likely dominate for the daytime HONO production in urban Beijing (Liu *et al.*, 2014)."

6.  *Page 10, line 317 to 319: The mean absolute error does not seem to be the best metric for evaluating the fitting of data that vary over more than one order of magnitude. This can lead to a bias where the high values are well represented but the relative deviation for the small values can be large. A better metric could be the average ratio (sum over all max(Vi_measured , Vi_fitted) / min(Vi_measured , Vi_fitted), where Vi is a data point for the considered parameter, e.g. [H₂SO₄]).*

Reply: In this study, the gaseous sulfuric acid concentrations are in a range of $(2.2\text{-}10.0) \times 10^6$ molecule cm$^{-3}$ in the 5-95% percentiles, whose variation is less than one order of magnitude. Nevertheless, we have now defined a metric of "relative error" (RE) to evaluate the fitting of data, which turns out to be 20.04 % (N1), 20.00 % (N2), 19.95 % (N3), 19.95 % (N4), 19.11 % (N5), 19.66 % (N6), and 19.34 % (N7), respectively. These results are consistent with our previous MAE results. The new metric is used throughout the revised manuscript and introduced as Eq. (8) (L293)

$$RE = \frac{1}{n} \cdot \sum_{i=1}^{n} \frac{|[H_2SO_4]_{proxy,i} - [H_2SO_4]_{meas.,i}|}{[H_2SO_4]_{meas.,i}} \qquad (8)$$

7.  *Page 11, line 346/347: The improvement relates to an increase of R from 0.85 to 0.86; this does not justify the word "significantly"; "improved slightly" instead of "improved significantly" is more appropriate.*

Reply: We have revised our manuscript, which reads (L400-L402) "When both [O₃] and [HONO] were introduced as OH precursors in the proxies N5-N7, REs have noticeable improvements, and correlation coefficients improved slightly."

8.  *Page 13, line 396: Aren't the different values of CS taken into account in the calculation, or do the authors mean that the different exponents for CS make the difference? What is the exponent for the CS in the Petäjä et al. study?*

Reply: We think it is the value of $CS^c$ in the proxies that makes the difference. For example, if the exponent C is very near to zero, then no matter how CS changes, the value of $CS^c$ would always be very close to 1, which means that this term would not influence the proxies at all. The CS in the Petäjä *et al.* study did not have an exponent.

9.  *Page 14, line 439: How can the proxies be evaluated for past measurements? If measurements exist for both sulfuric acid and the proxy parameters, these should be included in the present study.*

Reply: We failed to obtain previous data sets that include both gaseous sulfuric acid concentrations and other proxy parameters. As far as we know, there are two studies that measured gaseous sulfuric acid concentrations in Beijing (Zheng *et al.*, 2011; Cai *et al.*, 2017), but other key inputs for the proxy are not available from the two studies.

10. *Page 14, line 440: The importance of heterogeneous chemistry cannot be concluded from the presented data; this statement should be removed or supported with further data.*

Reply: We have removed this statement.

11. *Section 5: Discussion on the expected seasonality of the proxy-H2SO4 relation should be discussed. Very likely, the exponents can have different values for other meteorological condition, e.g., when desert dust contributes to the particle burden. In such a case, the exponent for the CS could become negative. This possibility should be mentioned/discussed.*

Reply: We have revised the conclusion section, and expanded the discussion on the applicability of the proxies in this study, which reads (L491-L496) "Furthermore, the proxies might be site-specific and season-specific. Since the proxies were derived with atmospheric parameters in winter, in urban Beijing, the exponents for atmospheric variables in the proxy could have different values for other cities or other seasons. Thus, the proxies in this study should be further tested before their application to other Chinese megacities or other seasons".

12. *Table 1: The values for ozone are quite low. Are these low values typical for wintertime conditions in Beijing?*

Reply: The ozone concentration in winter 2018 in Beijing is actually lower than those in the past years. In addition, as our station is not far from a traffic-heavy road, sometimes, $O_3$ could be completely diminished by NO.

13. *Table 3: What is the definition of the scaling factor k0? What are its units?*

Reply: The scaling factor $k_0$, which scales the calculated values from the proxy variables to match the measured sulfuric acid concentrations, is derived from the ratio of measured sulfuric acid concentrations and the proxy concentrations (without $k_0$ itself). The units of $k_0$ in different proxies are different, but together with units of all variables would result in a unit of molecule $cm^{-3}$.

14. *Figure 3: The yellow areas are hardly visible; it would probably be better to use colored lines instead of filled areas for this figure. In addition, the shape of the curves suggests a pronounced bias (the high values are on average underestimated, while the low values are overestimated). This bias can also be seen in the SI Figures. Using a different metric for the fitting (ratios instead of absolute differences, see above) could solve this issue.*

Reply: We have revised Figure 3. Colored lines are now used to present the performance of Proxy N2 and Proxy N7. A new metric of "relative error" is used to evaluate the fitting quality.

**Language comments:**

1. *Page 2, line 46: "less than" rather than "up to"?*

Reply: we have revised our manuscript, which reads (L49) "the relative errors were reduced up to 20 %".

2. *Page 3, line 57: "involve" instead of "demand"*

Reply: We have revised our manuscript accordingly, which reads (L61) "…involve the participation of gaseous sulfuric acid molecules".

3. *Page 3, line 73/74: delete "using a passive CIMS"*

Reply: We have removed this expression.

4. *Page 3, line 76: delete "associated equilibrium or"*

Reply: We have removed this expression.

5. *Page 3, line 79: replace "will be" with "are"*

Reply: We have revised our manuscript accordingly, which reads (L83-L84) "…known concentrations of OH radicals that are titrated into gaseous sulfuric acid".

6. *Page 4, line 90: replace "supposing" with "assuming"*

Reply: We have revised our manuscript accordingly, which reads (L95-L96) "assuming that gaseous sulfuric acid is formed from reactions between $SO_2$ and OH radicals".

*7.   Page 4, line 102: delete ", a proxy for condensational sink"*

We have removed this expression.

*8.   Page 4, line 108: delete "that characterized"*

We have removed this expression.

*9.   Page 4, line 114: replace "between" with "in"*

Reply: We have revised our manuscript accordingly, which reads (L120-L122) "…measured gaseous sulfuric acid concentrations are relatively similar in these environments…".

*10.  Page 6, line 156: "the sample flow"*

Reply: We have revised our manuscript accordingly, which reads (L163) "…the middle of the sample flow".

*11.  Page 6, line 177: "dependent on" instead of "that could be defined as a function of"*

Reply: We have revised our manuscript accordingly, which reads (L211-L212) "… a transition-regime correction factor dependent on the Knudsen number …".

*12.  Page 8, line 230: What is meant by the symbol xi'?*

Reply: We have revised our manuscript accordingly to make it clear, which reads (L274-L275) "… where $x_i$ can be an atmospheric variable such as UVB and $[SO_2]$".

*13.   Page 9, line 255: "in the 5-95% percentile range" instead of "was in . . . range of"*

Reply: We have revised our manuscript accordingly, which reads (L306) "in the 5-95 % percentile range, relatively similar to…".

*14.  Page 9, line 280: delete "an"*

Reply: We have removed this word.

*15.  Page 10, line 290: "lamination" does not seem to be the right word here, maybe better to use "layering with"*

Reply: We have revised our manuscript accordingly, which reads (L345) "A good correlation layering with $[SO_2]$ is evident…".

*16. Page 10, line 305: "has" instead of "have"*

Reply: We have revised our manuscript accordingly, which reads (L360) "…that has the UVB intensity and $[SO_2]$ …".

*17. Page 11, line 352: either delete "are tended to" or use "tend to"*

Reply: We have revised our manuscript accordingly, which reads (L408) "[HONO] and $[NO_x]$ tend to correlate linearly with each other …".

*18. Page 12, line 382: Do the authors mean "deviations" instead of "derivations"?*

Reply: Following this reviewer's comments #6 and #14, we have used a new metric of "relative error" (L293) instead of relative deviation in Figure 3.

*19. Page 13, line 390: "when averaged to a time resolution of 5 min" instead of "with a time resolution of 5 min"*

Reply: We have revised our manuscript accordingly, which reads (L437) "when averaged to a time resolution of 5 min".

*20. Page 13, line 411: Please provide the units for the parameters in this equation.*

Reply: We have added the units for the parameters, which reads (L457-L458) "The units of $[H_2SO_4]$ and $[SO_2]$ are molecule $cm^{-3}$, and the unite of UVB is W $m^{-2}$".

*21. Page 14, line 418/419: Please provide the units for the parameters in this equation.*

Reply: We have added the units for the parameters, which reads (L465-L466) "The units of $[H_2SO_4]$, $[SO_2]$, $[O_3]$ and [HONO] are molecule $cm^{-3}$, the unite of UVB is W $m^{-2}$, and the unite of CS is $s^{-1}$".

*22. Page 14, line 425/426 Please provide the units for the parameters in this equation.*

Reply: We have added the units for the parameters, which reads (L473-L474) "The unites of $[H_2SO_4]$, $[SO_2]$, $[O_3]$ and $[NO_x]$ are molecule $cm^{-3}$, the unite of UVB is W $m^{-2}$, and the unite of CS is $s^{-1}$".

*23. Page 14, 433/434: Please provide also the equation numbers and not just the proxy numbers (N2 and N7).*

Reply: We have now provided both the equation numbers and proxy numbers, which reads (L484-L485) "As a summary, we recommend using the simplest proxy (proxy N2 as shown in Eq. (9)) and a more accurate proxy (Proxy N7 as shown in Eq. (11)) for calculating daytime gaseous sulfuric acid …".

Reference

Cai, R., Yang, D., Fu, Y., Wang, X., Li, X., Ma, Y., Hao, J., Zheng, J. and Jiang, J. (2017) 'Aerosol surface area concentration: A governing factor in new particle formation in Beijing', *Atmospheric Chemistry and Physics*, 17(20), pp. 12327–12340. doi: 10.5194/acp-17-12327-2017.

[revised manuscript text omitted]

---

## Author Comment (AC2) · 30 Jan 2019

**RE: A point-to-point response to reviewers' comments**

"A proxy for atmospheric daytime gaseous sulfuric acid concentration in urban Beijing" (acp-2018-1132) by Yiqun Lu, Chao Yan, Yueyun Fu, Yan Chen, Yiliang Liu, Gan Yang, Yuwei Wang, Federico Bianchi, Biwu Chu, Ying Zhou, Rujing Yin, Rima Baalbaki, Olga Garmash, Chenjuan Deng, Weigang Wang, Yongchun Liu, Tuukka Petäjä, Veli-Matti Kerminen, Jingkun Jiang, Markku Kulmala, Lin Wang

We are grateful to the helpful comments from this anonymous referee, and have carefully revised our manuscript accordingly. A point-to-point response to the comments, which are repeated in italic, is given below.

In addition to the reviewers' comments, we have noticed and corrected a key typo from our previous version of manuscript. "The [NO$_2$] concentration" in our manuscript is in fact "the [NO$_x$] concentration". Correction of this term does not lead to changes in our conclusions.

**Reviewer #1's comments:**

*This study examines the relationship of [SO2] to [H2SO4] as a function of light intensity, particle concentration, and other gas phase reactants in Beijing. As the authors point out, this relationship is likely different in Beijing than in European and US cities. Overall, this study is straight forward and generally useful for research conducted in megacities. However, several issues should be address before this study can be considered for publication in ACP.*

Reply: We are very grateful to the positive viewing of our manuscript by Reviewer #1, and have now revised our manuscript accordingly.

**Major comments:**

1. *P5 147: Sulfuric acid concentration was measured using nitrate LToF-CIMS. It would be useful for the reader to know more details on how the sulfuric acid concentration was determined from the signals of the instrument. Does this measurement include sulfuric acid in molecular clusters (i.e. is fragmentation contributing to the sulfuric acid signal?) What are the estimated uncertainties of the sulfuric acid measurement? How do these uncertainties compare to the model predicted amounts?*

Reply: We measured signals of charged sulfuric acid as HSO$_4^-$ , HSO$_4^-$ HNO$_3$, and HSO$_4^-$ (HNO$_3$)$_2$, and charged clusters of HSO$_4^-$ H$_2$SO$_4$. The clusters of HSO$_4^-$ H$_2$SO$_4$ come from ion-induced clustering of neutral sulfuric acid and bisulfate ions within the LTOF-CIMS ion reaction zone, and also from the evaporation of dimethylamine (DMA) and the replacement of one molecule of H$_2$SO$_4$ with one bisulfate ion, HSO$_4^-$, during the NO$_3^-$ reagent ion charging of a stabilized neutral sulfuric acid dimer in the real atmosphere in presence of DMA or a molecule that works in the same way as DMA. Although the total signals of $HSO_4^-$, $HSO_4^- \cdot HNO_3$, and $HSO_4^- \cdot (HNO_3)_2$ were normally orders higher than that of $HSO_4^- \cdot H_2SO_4$, the sum of $HSO_4^-$, $HSO_4^- \cdot HNO_3$, $HSO_4^- \cdot (HNO_3)_2$ and $HSO_4^- \cdot H_2SO_4$ has been used to derive the gaseous sulfuric acid concentration.

The uncertainty in measured concentrations of sulfuric acid is estimated to vary between 21% and 51%, which is comparable with that in Kürten *et al.* (2012).

The uncertainties of proxies came from both those of CIMS and those of proxy methods. We have elaborated how the sulfuric acid concentration was determined in our revised manuscript, which reads (L164-L183), "For example, the atmospheric $H_2SO_4$ molecules would be charged by nitrate reagent ion $NO_3^-(HNO_3)_{0-2}$ and mainly produce $HSO_4^-$ ions (m/z = 96.9601 Th), $HSO_4^- \cdot HNO_3$ ions (m/z = 159.9557 Th), and $HSO_4^- \cdot (HNO_3)_2$ ions (m/z = 222.9514 Th). In addition, $HSO_4^- \cdot H_2SO_4$ ions (m/z = 194.9275 Th) were formed from ion-induced clustering of neutral sulfuric acid and bisulfate ions within the LToF-CIMS ion reaction zone, and also from the evaporation of dimethylamine (DMA) and the replacement of one molecule of $H_2SO_4$ with one bisulfate ion, $HSO_4^-$, during the $NO_3^-$ reagent ion charging of a stabilized neutral sulfuric acid dimer in the real atmosphere in presence of DMA or a molecule that works in the same way as DMA. During the campaign, the sample flow rate was kept at 8.8 slpm, since mass flow controllers fixed the sheath flow rate and the excess flow rate, and the flow into the mass spectrometer (around 0.8 slpm) was fixed by the size of a pinhole between the ionization source and the mass spectrometer. The concentration of gaseous sulfuric acid was then determined by Eq. (2).

$$[H_2SO_4] = \frac{HSO_4^- \cdot (HNO_3)_{0-2} + HSO_4^- \cdot H_2SO_4}{NO_3^-(HNO_3)_{0-2}} \cdot C \qquad (2)$$

where C is the calibration coefficient, and $NO_3^-(HNO_3)_{0-2}$, $HSO_4^- \cdot (HNO_3)_{0-2}$ and $HSO_4^- \cdot H_2SO_4$ represent the signals of corresponding ions and are in unites of counts per second (cps). The unit of resulting $[H_2SO_4]$ is molecule cm$^{-3}$ …".

*2. P6 162: Along these same lines, the authors comment that the calibration coefficient takes into account diffusion losses in the sampling line. Was this loss measured? It is a bit surprising that the calibration coefficient that Kurten et al. (2012) determined was 1.1 x10^10 cm-3 is the same in this study. I would have thought differences in instrument and sampling line losses (1.6 m is quite long) would have impacted this number. The authors should more clearly lay out how the sulfuric acid concentration was determined since it is an integral measurement for this paper.*

Reply: The loss rate was calculated assuming a diffusion loss of sulfuric acid in a circular tube under a laminar flow condition (Gormley and Kennedy, 1949). The identical values for the effective calibration coefficient are just by coincidence. Our calibration coefficient is $3.79 \times 10^9$ molecule cm$^{-3}$ in absence of the diffusion loss correction. We have elaborated our discussion on the calibration coefficient, which reads (L186-L190), "We obtain a calibration coefficient of $3.79 \times 10^9$ molecule cm$^{-3}$ for our instrument and use $1.1 \times 10^{10}$ molecule cm$^{-3}$ as the effective calibration coefficient, after taking into account the diffusion losses in the stainless-steel tube and the nitrate chemical ionization source".

*3. P6 L 176: The Fuchs-Sutugin transition kernel is used. There is associated error with using this kernel in the transition regime where sulfuric acid condenses on preexisting particles. Can the authors comment on this error? How sensitive is the fitting parameters to changes in the collision kernel? It may be helpful for the authors to use the empirically-derived collision kernel for the full regime from (Gopalakrishnan and Hogan Jr., 2011) to help limit the uncertainties from this parameter.*

Reply: We have calculated $H(Kn_D)$ from the nondimensionalized form of Fuchs-Sutugin and the First passage regression (Gopalakrishnan and Hogan Jr., 2011) in our $Kn_D$ range. The difference between the calculated $H(Kn_D)$ from two methods is within 8 %. Hence, we don't think calculation of the collision kernels in this study will lead to a significant uncertainty. Nevertheless, we have added this citation to give a more comprehensive discussion on the transition-regime correction factor, which reads (L211-L213), "$\beta_m$ represents a transition-regime correction factor dependent on the Knudsen number (Fuchs and Sutugin, 1971; Gopalakrishnan and Hogan Jr., 2011)".

*4. P5 134: The authors state that two months of measurements were conducted. It is not clear from the paper if all two months of measurements were used to determine the proxy relationship. Have the authors examined how the proxy relationship changes from day to day? Or week to week? The authors correctly state that the proxy relationship likely depends on location but does it also depend on time? It is possible that other processes that affect sulfuric acid concentrations (like Criegee intermediates) are not captured in the proxy relationship may play a larger role during some times of day than others.*

Reply: The intensive campaign was carried out from 9 February to 14 March, 2018 (L172), which lasts a bit more than a month. The performance of proxies could be different more or less on different days as the proxies are derived from a statistical analysis of the entire data set in this study. We have expanded the discussion about the applicability of the proxies, which reads (L491-L496) "Furthermore, the proxies might be site-specific and season-specific. Since we derived the proxies in winter in urban Beijing, the exponents of factors in the proxy for other cities or other seasons could have different values. Thus, the proxies in this study should be further tested before their application to other Chinese megacities or other seasons".

Criegee intermediates are not intended to be included in the proxy because Criegee intermediates seem not to play an important role in the daytime formation of sulfuric acid (L135-L139) (Boy et al., 2013; Mauldin et al., 2012).

*5. P13 line 389: If two months of measurements were taken, why was only one day used to compare measured to predicted sulfuric acid concentrations? How does the comparison for the other days look? It's not necessary to add graphs of these comparison, but a few lines stating the comparison for other days is necessary for the reader to determine how useful this proxy relationship is.*

Reply: The measurements lasted a bit more than a month as mentioned above. Figure 3 (now updated to a new version) presents a statistical comparison between measured and predicted sulfuric acid concentrations in all the measurement days. Relevant discussions can be found in section 4.4.

6. *P13 line 396: Authors state that the proxy relationship developed for the boreal forest and applied to Beijing is a factor of 20 too low due to differences in CS. Why didn't the authors use the Beijing CS values when applying Petäjä et al.'s proxy relationship? Would the differences between measured and predicted from Petäjä then be smaller?*

Reply: We actually used atmospheric variables including CS values from our measurements when applying Petäjä *et al.*'s proxy relationship. The reason for the poor performance of Petäjä *et al.*'s proxy relationship on Beijing data could be the much higher CS values in Beijing with a much more complex atmosphere.

7. *Figure 4: It would be useful for the reader to see timelines of all the measured concentrations that go into the proxy relationships as well.*

Reply: Here we present results from a comprehensive campaign with participation from multiple universities and institutions. As a result, this manuscript will only focus on the development of the statistical analysis of the sulfuric acid proxy, and other manuscripts in preparation will discuss the variations of atmospheric variables.

**Minor comments:**

1. *P1 Line 28: desirable for the atmospheric. . .*

Reply: We have revised our manuscript accordingly, which reads (L28) "…highly desirable for the atmospheric chemistry community".

2. *P1 36-27 change one of the "using"*

Reply: We have revised our manuscript accordingly, which reads (L37-L39) "A proxy for atmospheric daytime gaseous sulfuric acid concentration was derived through a statistical analysis method by using the UVB intensity, [$SO_2$], condensation sink (CS), [$O_3$], and [HONO] (or [$NO_x$]) as the predictor variables".

3. *P3 Line 57: sulfuric acid DMA system. The citation for Petäjä et al. (2011) might not be the best. Several studies have pointed out potential experimental issues with this study (Jen et al., 2014; Kürten et al., 2014).*

Reply: We have updated the citations, which reads (L59-L61) "…$H_2SO_4$-DMA-$H_2O$ ternary nucleation (Almeida et al., 2013; Jen et al., 2014; Kürten et al., 2014; Petäjä et al., 2011; Yao et al., 2018)".

4. *P3 line 57: demand participation is a strange phrase. Maybe necessitates participations?*

Reply: We have revised our manuscript, which reads (L61) "…involve the participation of gaseous sulfuric acid molecules".

5.  *P3 line 59: Would be worth reading and citing (Kuang et al., 2012) for sulfuric acid growth rates.*

Reply: We have added this citation, which reads (L61-L63) "In addition, the condensation of gaseous sulfuric acid onto newly-formed particles contributes to their initial growth (Kuang et al., 2012; Kulmala et al., 2013)".

6.  *P3 line 62: Knowing sulfuric acid concentrations prior to a nucleation event is also important.*

Reply: We have revised our manuscript, which reads (L63-L67)"Quantitative assessments of the contribution of gaseous sulfuric acid to both the new particle formation rates and the particle growth rates require real-time measurements of gaseous sulfuric acid concentrations prior to and during the NPF events (Nieminen et al., 2010; Paasonen et al., 2010) ".

7.  *P3 Line 68: NO3- and ligands.*

Reply: We have revised our manuscript, which reads (L72) "…with $NO_3^-$ and its ligands as reagent ions".

8.  *P3 line 68: CIMS is actually a pretty broad class of instruments. The low detection limit for sulfuric acid is because the instrument ionizes and samples at atmospheric pressure, which is different than the traditional CIMS.*

Reply: We have revised the sentence accordingly, which reads (L72-L73) "because nitrate CIMS with an atmospheric pressure interface (API) has a low detection limit …".

9.  *P3 line 80: (Chen et al., 2012) shows a nice figure of sulfuric acid concentrations measured at numerous locations around the world. Not critical to add the citation but worth taking a look at.*

Reply: We appreciate that this reviewer points out a very important paper presenting the measurements of sulfuric acid in different locations. We would like to add the citation, which reads (L84-L88) "Thereafter, measurements of sulfuric acid using CIMS have been performed around the world (e.g., Berresheim et al., 2000; Bianchi et al., 2016; Chen et al., 2012; Jokinen et al., 2012; Kuang et al., 2008; Kürten et al., 2014; Kurtén et al., 2011; Petäjä et al., 2009; Weber et al., 1997; Zheng et al., 2011) …".

10. *P3 line 83: has been proven*

Reply: We have revised our manuscript accordingly, which reads (L88) "CIMS has been proven to be a robust tool ...".

*11. P4 line 105: After reading this, the reader will naturally wonder why is there a positive correlation between CS and sulfuric acid concentration?*

Reply: We have revised our manuscript accordingly, which reads (L108-L111) "In several proxies developed by Mikkonen et al. (2011), the correlation between the gaseous sulfuric acid concentration and CS is positive, which is against what one would expect because a larger CS normally leads to a faster loss for gaseous sulfuric acid". In addition, we put the detailed discussion of this issue in section 4.2 and 4.3.

*12. P4 line 108: locations that characterize with an. . . one or two of those words are not correct.*

Reply: We have revised our manuscript, which reads (L114-L115) "…in locations with atmospheric environments different from those in the six sites of that study".

*13. P4 line 110: Please state the range of CS in addition to how much higher it is compared to other locations.*

Reply: We have stated the range of CS, which reads (L116-L117) "Beijing is a location with typical values of CS (*e.g.*, 0.01-0.24 $s^{-1}$ in the 5-95% percentiles in this study) being 10-100 times higher …".

*14. P4 line 113: For north America: how do these numbers compare to Mexico City?*

Reply: The level of $SO_2$ in Beijing has decreased significantly in recent years as we have presented in L488-L489. Nevertheless, the $SO_2$ concentration in Mexico City in 2003 is comparable with our measured $SO_2$ in Beijing. We have added one citation regarding the $SO_2$ measurements and two citations regarding the sulfuric acid concentration in Mexico City, which reads (L118-L123) "…typical $SO_2$ concentrations being 1-10 times higher (Wang et al., 2011a; Wu et al., 2017) than those in Europe and North America (Dunn et al., 2004; Mikkonen et al., 2011), yet measured gaseous sulfuric acid concentrations are relatively similar in these environments (Chen et al., 2012; Smith et al., 2008; Wang et al., 2011b; Zheng et al., 2011)."

*15. P5 119: OH radicals*

Reply: We have revised our manuscript, which reads (L126) "… a potentially important source of OH radicals in the atmosphere".

*16. P5 119: remove the not only and but also. It is harder to read with them there.*

Reply: We have removed the two expressions.

*17. P5 line 128: Criegee should be capitalized*

Reply: We have capitalized "C".

*18. P6 line 153: was guided through. . . strange phrasing*

Reply: We have revised our manuscript, which reads (L160) "…was introduced into a PhotoIonizer …".

*19. P6 line 154: Is this a custom-built inlet? If so, could the authors provide a diagram and*

*write in the dimensions?*

Reply: The inlet is a commercial product from Aerodyne Research, Inc.

*20. P6 line 160: CIMS was calibrated. How? It would be useful to describe this procedure in brief.*

Reply: We have added a brief introduction of the calibration process, which reads (L183-L186) "The CIMS was calibrated during the campaign with a home-made calibration box that can produce adjustable concentrations of gaseous sulfuric acid from $SO_2$ and OH radicals following the protocols in previous literatures (Kürten et al., 2012; Zheng et al., 2015)."

*21. P6 line 164: should it be ToFTools?*

Reply: The code maker just named it as tofTools.

*22. P6 line 166: 1 nm. Is this mobility diameter?*

Reply: Yes, it is mobility diameter.

*23. P7 line 213: Authors should better justify pseudo-steady state assumption*

Reply: The Mikkonen et al. (2011) study has indicated that the pseudo-steady state assumption holds well for typical atmospheric conditions. Furthermore, the errors for nonlinear proxies derived from the pseudo-steady state assumption in Mikkonen et al. study are in a range of 40-42 %, whereas ours are in a range of 17.6-19.2 % when evaluating the performances of the proxies with the "error" metric in Mikkonen et al. study. Therefore, we think that the pseudo-steady state assumption can be applied to our proxies.

*24. P8 paragraph starting on line 228: This was a difficult paragraph to understand. Can the authors better phrase it to explain the differences in parameters?*

Reply: We have revised the paragraph, which reads (L272-L283) "In practice, the exponents for variables in nonlinear fitting procedures are rarely equal to 1 (Mikkonen et al., 2011), so we replaced the factors $x_i$ with $x_i^{w_i}$ in the proxy, where $x_i$ can be an atmospheric variable and $w_i$ defines $x_i$' exponent in the proxy. Since $k$ is a temperature-dependent reaction constant and varies within a

% range in the atmosphere temperature range of 267.6 - 292.6 K, *i.e.*, the actual atmospheric temperature variation in this study, we approximately regard $k$ as a constant and use a new scaling factor $k_0$. This methodology has been used previously in the proxies of gaseous sulfuric acid in Hyytiälä, Southern Finland (Petäjä et al., 2009). As a result, the general proxy equation can be written as Eq. (7), with the UVB intensity, [$SO_2$], condensation sink (CS), [$O_3$], and [HONO] (or [$NO_x$]) as predictor variables:".

*25. P8 line 242: a matlab software. A custom-made one? Or just a function in matlab?*

Reply: The nonlinear curve-fitting procedures are performed by a custom-made MATLAB software. We have revised our manuscript, which reads (L287-L289) "The nonlinear curve-fitting procedures using iterative least square estimation for the proxies of gaseous sulfuric acid concentration based on Eq. (7) were performed by a custom-made MATLAB software."

*26. P9: 1-2 orders of magnitude. Maybe change to 10-100 times higher to be more clear.*

Reply: We have revised our manuscript, which reads (L303-L304) "which is about 10-100 times higher …".

*27. P9 line 261: 60% RH does not seem dry.*

Reply: The mean RH in this campaign is 28%. We have revised our manuscript, which reads (L312-L313) "In addition, Beijing is dry in winter with a mean ambient relative humidity of 28% during the campaign".

*28. P9 272: I do not understand how the correlation coefficient numbers are consistent with accepted formation pathways? Does the formation pathways have powers that are less than 1?*

Reply: We have rephrased the sentence, which reads (L324-L326) "…which indicate that [$SO_2$] and UVB have important influences on the formation of atmospheric gaseous sulfuric acid".

*29. P9 line 276: Authors should explain potential reasons why sulfuric acid positively correlates with CS.*

Reply: We discussed in P18 Line 108 that "In several proxies developed by Mikkonen et al. (2011), the correlation between the gaseous sulfuric acid concentration and CS is positive, which is against what one would expect because a larger CS normally leads to a faster loss for gaseous sulfuric acid". In this campaign, CS correlates well with [$SO_2$] ($r = 0.83$), which suggests that a high CS value could serve as an indicator of atmospheric particulate pollution, and be accompanied with a high concentration of $SO_2$ that is propitious for the formation of gaseous sulfuric acid. Please also refer to our discussion of this issue in section 4.3.

*30. P10 300: molecules cm-3 is normally written as just cm-3.*

Reply: the unit of molecule cm$^{-3}$ has been extensively used in the literature and we decide to keep this unit.

*31. P10 line 316: Authors mention that proxy relationship is location specific. Why then did the authors use the justification for not including RH based upon conclusions drawn from a different location?*

Reply: We made a test by introducing RH into the proxy N1 (containing CS terms) and resulted in a RH-corrected CS term (CS·RH) instead of CS as what Mikkonen *et al*. have done (2011). The performance of proxy has not significantly improved (REs changed from 20.04 % to 19.83 %, see our reply to comments #6 and #14 from Reviewer #2 for REs) and the exponent of CS was still close-to-zero (from 0.03 to -0.02). We have rephrased our discussion on RH, which reads (L371-L374) "…because a test by introducing RH into the proxies do not result in a significantly better performance, which is consistent with those conclusions in the Mikkonen et al. study (2011)."

*32. P11 line 324: "unlike assumed in Eq. (3)" wording seems incorrect*

Reply: We have revised the sentence, which reads (L379-L380) "…unlike the assumption in Eq. (6) …".

*33. P11 line 324: The naming convention between the equations in table 3 and the equations in the paper is confusing. Which equation 3 does this line refer to?*

Reply: We have used the term "function" in Table 3 to avoid confusion.

*34. Page 12 line 356: "Only occasionally slightly higher" too many adverbs. Rephrase*

Reply: We have revised the sentence, which reads (L412-L413) "Occasionally, higher [HONO]/[NO$_x$] ratios could be seen in the morning".

*35. Page 12 line 356: The authors refer to a previous study to justify linearity of NO2 and HONO. Where was the location of this study? This paragraph is general is difficult to discern results from previous studies and results from this study. Please make this more clear.*

Reply: We have revised this paragraph to focus on the measurements in Beijing, which reads (L406-L420) "Although so far the proxy N5 had the best fitting quality, it is impractical to explicitly include [HONO] because HONO measurements are very challenging. As shown in Fig. 2, [HONO] and [NO$_x$] tend to correlate linearly with each other in the daytime during this campaign, with a linearly fitted [HONO]/[NO$_x$] ratio of around 0.03 and a relative error of 0.42. Occasionally, higher [HONO]/[NO$_x$] ratios could be seen in the morning, which might be due to the fact that HONO concentration could have an accumulation process during the nighttime and lead to a deviation from the steady state. Therefore, due to the good correlation, the proxy N7 replaces [HONO] by [NO$_x$], a more easily measured variable, and performs equally well with the proxy N5."

*36. Page 12 line 376: authors should specific that this cover sulfuric acid concentrations for this location. 10ˆ6 cm-3 does not cover sulfuric acid concentrations around the world.*

Reply: We have revised the sentence, which reads (L425-L427) "…in the sulfuric acid concentration range of $(2.2 - 10) \times 10^6$ cm$^{-3}$, which covers the 5-95% percentiles of sulfuric acid concentrations in this study."

*37. Page 13 line 416: It is a bit confusing that the authors mention that proxy N5 is the most accurate when they spend most of the paper justifying the use of N7. Maybe change the wording "for the best proxy accuracy" or consider rewording this section to make it a bit less confusing/*

Reply: We have revised the sentence, which reads (L463) "For a comprehensive consideration of the formation pathways of OH radicals…".

*38. Page 14 line 439: I do not understand how this work has shown the importance of heterogenous chemistry as a potential source of OH. Was this mentioned somewhere else in the main paper? The authors should better justify this point if they want to include in the summary.*

Reply: We have removed this statement.

*39. Figure 1-2: What day were these measurements done?*

Reply: Figure 1-2 show all the measured data points during the campaign from 9 February to 14 March, 2018. We have included the duration of measurements in the revised figure caption.

*40. Figure 2: Can the authors explain why there seems to be clear break up group of points during the early morning that do not follow the linear trend?*

Reply: We have expanded the discussion, which reads (L412-L415) "Occasionally, higher [HONO]/[NO$_x$] ratios could be seen in the morning, which might be due to the fact that HONO concentration could have an accumulation process during the nighttime and lead to a deviation from the steady state."

*41. Figure 4: As mentioned above, it would be useful to show the time lines for the other measured concentrations (CS, OH, NO2, etc.) that the proxy model uses.*

Reply: Here we present results from a comprehensive campaign with participation from multiple universities and institutions. As a result, this manuscript will only focus on the development of the statistical analysis of the sulfuric acid proxy, and other manuscripts in preparation will discuss the variations of atmospheric variables.

Reference

[revised manuscript text omitted]

---

## Author Comment (AC3) · 30 Jan 2019

**RE: A point-to-point response to reviewers' comments**

"A proxy for atmospheric daytime gaseous sulfuric acid concentration in urban Beijing" (acp-2018-1132) by Yiqun Lu, Chao Yan, Yueyun Fu, Yan Chen, Yiliang Liu, Gan Yang, Yuwei Wang, Federico Bianchi, Biwu Chu, Ying Zhou, Rujing Yin, Rima Baalbaki, Olga Garmash, Chenjuan Deng, Weigang Wang, Yongchun Liu, Tuukka Petäjä, Veli-Matti Kerminen, Jingkun Jiang, Markku Kulmala, Lin Wang

We are grateful to the helpful comments from Dr. Santtu Mikkonen, and have carefully revised our manuscript accordingly. A point-to-point response to the comments, which are repeated in italic, is given below.

In addition to the reviewers' comments, we have noticed and corrected a key typo from our previous version of manuscript. "The [$NO_2$] concentration" in our manuscript is in fact "the [$NO_x$] concentration". Correction of this term does not lead to changes in our conclusions.

**Santtu Mikkonen's comments:**

*It is interesting to see how the sulphuric acid concentration can be approximated in highly polluted environment, as we did not have such data when we were making our paper Mikkonen et al. (2011). Even more interesting is, that your recommended proxy N2 is quite close to our second recommendation, simple proxy L3, having SO2 power to 0.5 when you have power of 0.4. In addition, I was surprised that the H2SO4 concentration was not higher than shown in Table 1. We had similar average concentrations in San Pietro Capofiume and considerably higher in Atlanta, even though they are less polluted environments. Could you add a comment on that?*

Reply: We are very grateful to the positive viewing of our manuscript by Dr. Santtu Mikkonen, and have now revised our manuscript accordingly.

As Dr. Mikkonen has noticed, Beijing did not in this campaign have a higher average concentration of sulfuric acid than other cities, which can be potentially explained by the fact that, firstly, the averaged condensation sink in Beijing in this campaign is around 0.11 s$^{-1}$ that corresponds to a very efficient removal of gaseous sulfuric acid, and secondly, the $SO_2$ concentration has dramatic reduced in recent years in Beijing as we have mentioned in L488-L489.

*I just want to ask about Figure 4: Why only one day, and not averages over whole period such that uncertainty would also be indicated, is shown in the figure? In addition, why comparison only to Boreal forest-proxy from Petäjä et al, why not to Mikkonen et al., who had data from multiple sites?*

Reply: Figure 3 (now updated to a new version) presents a statistical comparison between measured and predicted sulfuric acid concentrations over the whole period.

We compared our results with the Petäjä *et al.* study instead of the Mikkonen *et al.* study, simply because we only measured the UVB and a correlation between UVB and the global radiation cannot be established.

*A minor comment on the use of p-value as a screening factor for correlation (in line 266): it is not recommended. See e.g. Greenland et al. (2016): DOI 10.1007/s10654- 016-0149-3*

Reply: We removed p-values as a screening factor for correlations. Now all the correlations are shown in Table 2.

[revised manuscript text omitted]